# RoboScape: Physics-informed Embodied World Model

**Yu Shang**[1], **Xin Zhang**[2], **Yinzhou Tang**[1], **Lei Jin**[1], **Chen Gao**[1], **Wei Wu**[2]*, **Yong Li**[1]*

[1]Tsinghua University
[2]Manifold AI

## Abstract

World models have become indispensable tools for embodied intelligence, serving as powerful simulators capable of generating realistic robotic videos while addressing critical data scarcity challenges. However, current embodied world models exhibit limited physical awareness, particularly in modeling 3D geometry and motion dynamics, resulting in unrealistic video generation for contact-rich robotic scenarios. In this paper, we present RoboScape, a unified physics-informed world model that jointly learns RGB video generation and physics knowledge within an integrated framework. We introduce two key physics-informed joint training tasks: temporal depth prediction that enhances 3D geometric consistency in video rendering, and keypoint dynamics learning that implicitly encodes physical properties (e.g., object shape and material characteristics) while improving complex motion modeling. Extensive experiments demonstrate that RoboScape generates videos with superior visual fidelity and physical plausibility across diverse robotic scenarios. We further validate its practical utility through downstream applications including robotic policy training with generated data and policy evaluation. Our work provides new insights for building efficient physics-informed world models to advance embodied intelligence research. Our code and demos are available at: https://github.com/tsinghua-fib-lab/RoboScape.

## 1 Introduction

The advancement of large language and vision models [1, 2] has demonstrated the critical role of high-quality, large-scale training data for their superior performance. However, the robotic learning is significantly hindered by the prohibitive cost of collecting real-world data [3, 4, 5, 6], which often relies on human teleoperation to acquire high-quality demonstrations. This limitation poses a great challenge for scaling robotic learning and deploying agents in complex, real-world environments.

World models [7, 8, 9, 10, 11], which simulate environmental dynamics by predicting future states based on current observations and given actions, offer a promising solution to this data scarcity problem. Such models hold significant promise for advancing embodied intelligence by generating realistic robotic data [12] and enabling scalable simulation environments [13]. However, current embodied world models [13, 14, 15] predominantly focus on video generation, with training objectives centered on optimizing the RGB pixels. While capable of producing visually plausible 2D images, they often fail to maintain crucial physical properties, such as motion plausibility and spatial consistency [16]. Particularly, in robotic manipulation tasks involving deformable objects (e.g., cloth), the generated videos frequently contain artifacts such as unrealistic object morphing or discontinuous motion. These limitations become particularly detrimental in interaction-rich robotic scenarios, where even minor physical inconsistencies can dramatically compromise the effectiveness of learned policies.

---

*Corresponding author, correspondence to liyong07@tsinghua.edu.cn.

39th Conference on Neural Information Processing Systems (NeurIPS 2025).

The root cause lies in existing models' overreliance on visual token fitting without awareness of physical knowledge [17, 18, 19]. To address this, we propose a physics-informed world model that jointly learns depth information and temporal keypoint consistency to implicitly encode physical constraints. Existing efforts of integrating physical knowledge into video generation fall into three categories: physics-prior regularization, physics simulator-based knowledge distillation, and material field modeling. Current regularization-based methods enforce constraints such as local rigidity [20] or rotational similarity [21] on Gaussian splatting (GS) features or 3D point clouds. However, these methods are limited to narrow domains like human motion [22] or rigid-body dynamics [20], hindering generalization to diverse robotic scenarios. Another line of work employs physics simulators to extract motion signals or semantic maps as conditions to guide video generation models [23, 24, 25, 26]. Although this approach yields reliable physical priors, the resulting cascaded pipeline introduces excessive computational complexity, hindering their practical deployment. There have been some recent works trying to enhance the physical simulation via material field modeling [27, 28]. However, such methods are confined to object-level modeling and are hard to apply to scene-level generation.

To overcome these limitations, we propose RoboScape, a physics-informed world model based on a multi-task learning auto-regressive framework to generate visually realistic and physics-adherent robotic videos. Specifically, our approach incorporates physics knowledge through two auxiliary physics-informed supervision tasks within the world model itself to alleviate heavy external model cascading. First, to empower the model with 3D spatial physical understanding, we augment the RGB prediction backbone with a temporal depth prediction branch and inject the learned depth features into the RGB prediction to enhance spatial awareness. Such synergistic learning of temporal depth maps enables the model to implicitly acquire 3D scene reconstruction priors rather than merely fitting 2D RGB images. Second, we introduce an adaptive keypoint dynamics learning task to address unrealistic object deformation and implausible motion issues. To achieve this, we first perform dynamic keypoint sampling to automatically identify regions with significant motion (typically involving robots and interacting objects), then encourage temporal token consistency for these keypoints across frames. Through this, the model effectively captures the deformation properties and motion behaviors of objects, implicitly encoding material properties (e.g., rigidity and softness) through self-supervised keypoint consistency, eliminating the need for explicit material modeling. Although some recent world models [29, 30] also explore joint RGB-depth prediction, their learning remains constrained at the image level, failing to capture the fine-grained motion dynamics and object deformation details that are crucial for robotic manipulation scenarios. Furthermore, these approaches exhibit a performance trade-off, where gains in 3D perception come at the cost of reduced RGB prediction fidelity. Differently, our model captures global spatial knowledge through learning temporal depth dynamics, while modeling local object deformation and motion characteristics via learning temporal keypoint tracking.

We conduct comprehensive experiments to evaluate our world model from three aspects: video generation quality, robotic policy learning using synthetic data, and robotic policy evaluation. RoboScape achieves state-of-the-art performance in both RGB and depth prediction accuracy, achieving a superior balance between these metrics compared to existing world model baselines. Additionally, we validated that synthetic data from our world model consistently improves the performance of robotic policy models within a simulated robotic environment including Diffusion Policy [31] and pi0 [32], confirming the model's practical utility for robotic learning. Finally, our model can also serve as a reliable policy evaluator, with assessment results showing strong correlation with ground-truth simulator outcomes, confirming our model's capability to accurately model the physical world.

In summary, the main contributions of the paper are as follows:

- We propose RoboScape, a physics-informed embodied world model that unifies RGB video generation, temporal depth prediction, and adaptive keypoint tracking in a joint learning framework, achieving both high visual fidelity and physical plausibility.

- We design an automated robotic data processing pipeline with physical prior information labels. Trained on the carefully curated large-scale, high-quality dataset, our model achieves SOTA performance on visual quality, geometric accuracy, and action controllability.

- We demonstrate the practical utility of RoboScape on downstream applications including robotic policy training and evaluation. Extensive experimental results demonstrate its effectiveness in accurately modeling embodied environments, validating its potential for advancing real-world robotic deployment.

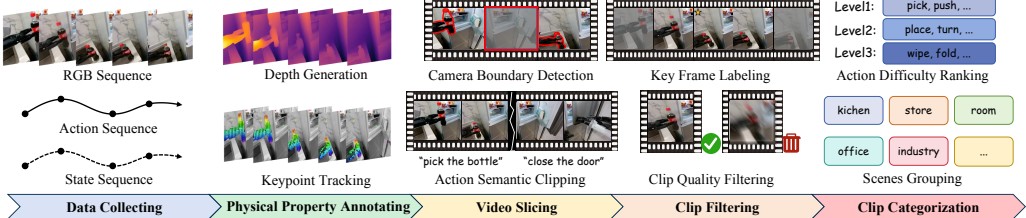

Figure 1: Illustration of the proposed robotic data processing pipeline with physical priors annotation.

## 2 Methodology

### 2.1 Problem Formulation

In this work, we focus on robot manipulation scenarios and learn an embodied world model $f_\theta$ as a dynamics function that predicts the next visual observation $\mathbf{o}_{t+1}$ given past observations $\mathbf{o}_{1:t}$ and robotic actions $\mathbf{a}_{1:t}$:

$$\mathbf{o}_{t+1} \sim f_\theta(\mathbf{o}_{t+1}|\mathbf{o}_{1:t}, \mathbf{a}_{1:t}), \tag{1}$$

where $\mathbf{o} \in \mathbb{R}^{H \times W \times 3}$ is a video frame and $\mathbf{a} \in \mathbb{R}^k$ is a $k$-degree continuous action control vector.

### 2.2 Robotic Data Processing Pipeline with Physical Priors Annotation

Learning a physics-informed embodied world model requires high-quality dataset covering high-resolution RGB and depth sequences, action sequences that control the robot, and state sequences that the robot executes. In this section, we present our data processing pipeline to construct a multi-modal embodied dataset with physical priors based on AGIBOT-World dataset [6], as shown in Fig. 1.

**Physical Property Annotating based on Depth Generation and Keypoint Tracking.** Integrating explicit physics constraints remains a significant challenge for current video generation-based world models. To address this, our approach concentrates on two crucial, visually-expressible physics constraints highly relevant to robotic manipulation: temporal depth consistency and keypoint motion trajectories. These features can be efficiently extracted using off-the-shelf pretrained models, enabling enhanced generalization while maintaining practical feasibility. Specifically, we utilize Video Depth Anything [33] to generate the depth map sequence of the video. Furthermore, we apply SpatialTracker [34] as the keypoint tracking model to sample the keypoint and track their trajectories.

**Video Slicing based on Camera Boundary Detection and Action Semantic.** The original videos have different attributes, such as lengths and resolution, with camera jumps or editing traces, and a video may contain multiple action semantics. Thus, we slice the video into clips with normalized attributes, consistent motion, no camera jumps, and single action semantics. Specifically, we use TransNetV2 [35] to perform camera boundary detection and use Intern-VL [36] to generate the action semantic of a specific clip.

**Clip Filtering based on Key Frame and Clip Quality.** The generated clips are highly heterogeneous in terms of quality, semantics, and presentation form. To ensure the validity and adaptability of the training data, we introduce a clip filtering mechanism including: (1) using FlowNet [37] to filter out clips with indistinct motion and disordered movement patterns, and (2) using Intern-VL [36] to label the key frame of the clip and filter out the frames without explicit relationship to the key frame.

**Clip Categorization based on Action Difficulty and Scenes.** In this stage, we categorize and reorganize the dataset based on action difficulty and clip scenes to support the curriculum learning strategy [38], which trains the world model from easier to harder tasks.

### 2.3 RoboScape: A Physics-informed Embodied World Model

RoboScape is designed to achieve frame-level action-controllable robot video generation, enabling interactive future frame prediction. At its core, we adopt an auto-regressive Transformer-based framework that iteratively predicts the next frame based on historical frames and the current robot

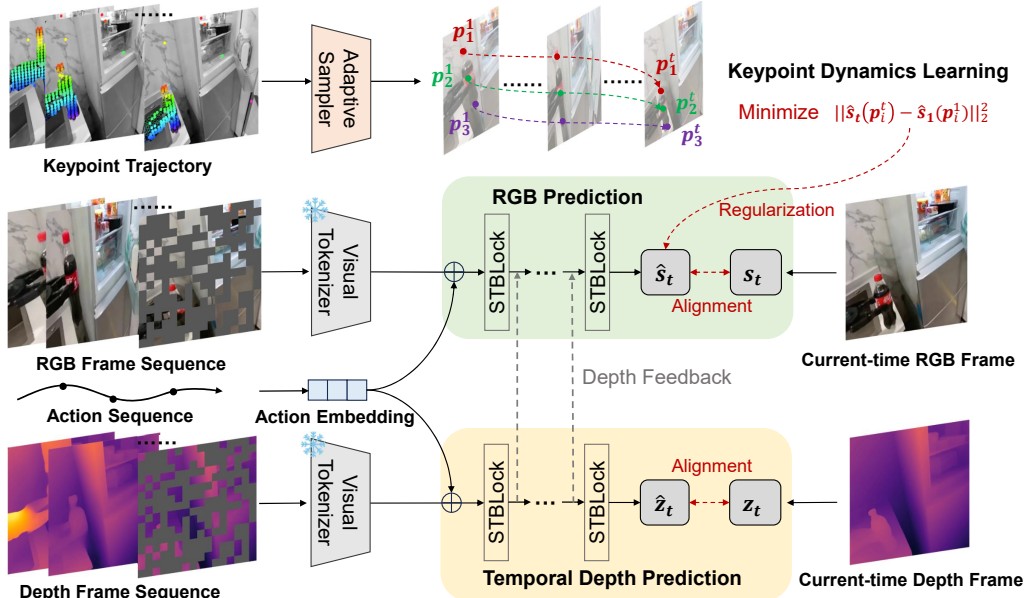

Figure 2: Overview of the physics-informed world model, where physical knowledge is integrated through joint learning of temporal depth estimation and adaptively sampled keypoint dynamics.

action. To enhance the physical plausibility of generated videos, we introduce two physics-informed auxiliary training tasks in addition to the normal RGB image prediction: (1) temporal depth prediction, which encourages global geometric consistency across frames, and (2) adaptively sampled keypoint dynamics learning, which captures the motion and deformation details of local dynamic objects. The whole pipeline is illustrated in Figure 2. Joint training with these physics-aware regularizers provides an efficient approach to embed physical priors into world models, significantly reducing the reality gap between generated videos and real-world dynamics.

**Video Tokenization.** To enable efficient video generation, we leverage MAGVIT-2 [39] to compress raw RGB frames $\mathbf{o}_{1:T} \in \mathbb{R}^{T \times H \times W \times 3}$ into discrete latent tokens $\mathbf{s}_{1:T} \in \mathbb{R}^{T \times H' \times W' \times D}$, where $H' = H/\alpha$ and $W' = W/\alpha$ denote the reduced spatial dimensions ($\alpha$ being the downsampling factor), and $D$ represents the latent channel dimension. Similarly, we tokenize temporal depth maps $\mathbf{d}_{1:T} \in \mathbb{R}^{T \times H \times W \times 1}$ into latent depth tokens $\mathbf{z}_{1:T} \in \mathbb{R}^{T \times H' \times W' \times D}$.

**Geometry Consistency Enhancement via Temporal Depth Prediction.** While RGB-based video generation has achieved remarkable progress, it often suffers from inconsistent 3D geometry due to the lack of explicit spatial constraints. Considering that inter-frame depth variations encode crucial 3D structure information, we propose to jointly learn temporal RGB and depth information, leveraging depth features as geometric constraints to ensure spatially coherent video generation. For uniform modeling, we also utilize MAGVIT-2 as the visual encoder for depth maps. We convert depth maps to a three-channel RGB format to ensure compatibility with MAGVIT-2. For joint prediction of both RGB and depth images, we propose a dual-branch co-autoregressive Transformer (DCT). Each branch consists of a stack of 32 spatial-temporal Transformer (ST-Transformer) blocks, which implement a causal attention mechanism in the temporal attention layers for generation causality, and bidirectional attention in the spatial attention layers to enable full context modeling.

At timestep $t$, the model processes historical latent tokens through parallel branches $\mathcal{F}_{\text{RGB}}$ and $\mathcal{F}_{\text{Depth}}$, conditioned on learned action embeddings $\mathbf{c}_{1:t-1} \in \mathbb{R}^{(t-1) \times 1 \times 1 \times D} : \mathcal{E}_a(\mathbf{a}_{1:t-1})$ and position embeddings $\mathbf{e}_{1:t-1} \in \mathbb{R}^{(t-1) \times H' \times W' \times D}$, where $\mathcal{E}_a$ denotes the robot action encoder. The auto-regressive prediction of each branch is formulated as:

$$\begin{aligned} \hat{\mathbf{s}}_t &= \mathcal{F}_{\text{RGB}}(\mathbf{s}_{1:t-1} \oplus \mathbf{c}_{1:t-1} \oplus \mathbf{e}_{1:t-1}), \\ \hat{\mathbf{z}}_t &= \mathcal{F}_{\text{Depth}}(\mathbf{z}_{1:t-1} \oplus \mathbf{c}_{1:t-1} \oplus \mathbf{e}_{1:t-1}), \end{aligned} \tag{2}$$

where $\oplus$ denotes element-wise addition with broadcasting. Empirically, we find that simple additive fusion provides effective action control while maintaining model efficiency.

To inject depth predictions as physical priors into the RGB branch and enhance spatial structure fidelity of rendered videos, we introduce cross-branch interaction pathways. Specifically, at each ST-Transformer block $l$, we project the depth branch's intermediate features $\mathbf{h}_{\text{depth}}^l$ and fuse them additively with the corresponding RGB features:

$$\mathbf{h}_{\text{RGB}}^l = \mathbf{h}_{\text{RGB}}^l + \mathcal{W}^l(\mathbf{h}_{\text{depth}}^l), \tag{3}$$

where $\mathcal{W}^l$ is a learnable linear projection layer. This hierarchical feature fusion enables the RGB branch to maintain precise geometric structure while generating photorealistic video frames. Both RGB and depth branches are optimized using the cross-entropy loss of tokens:

$$\mathcal{L}_{\text{RGB}} = -\sum_{t=1}^T \mathbf{s}_t \log p(\hat{\mathbf{s}}_t), \quad \mathcal{L}_{\text{Depth}} = -\sum_{t=1}^T \mathbf{z}_t \log p(\hat{\mathbf{z}}_t). \tag{4}$$

**Implicit Material Understanding via Keypoint Dynamics Learning.** Modeling physically plausible object deformations and motions in robot manipulation scenarios remains challenging for RGB-based world models, as material properties (e.g., rigidity, elasticity) cannot be effectively learned through RGB pixel fitting alone. While physics engines provide accurate simulations, their computational expense and scene-specific constraints limit practical applicability. To tackle this, we propose a keypoint-induced material learning approach, with the insight that physical material understanding can emerge from self-supervised tracking of contact-driven keypoint dynamics. For example, when a robot places an apple into a plastic bag, accurately capturing the motion of keypoints on the deforming bag implicitly captures the material properties. This method can be integrated naturally with video generation frameworks while maintaining strong generalization capabilities.

Specifically, for each video $\mathcal{V}$, we utilize SpatialTracker [34] to densely sample $N_0$ keypoints in the initial frame and track their temporal coordinate trajectories across $T$ frames, yielding $\mathcal{T}_{dense} = \{(\mathbf{p}_i^1, ..., \mathbf{p}_i^T)\}_{i=1}^{N_0}$, where the element $\mathbf{p}_i^t \in \mathbb{R}^2$ represents its coordinates in the tokenized feature map of frame $t$. Rather than relying on costly segmentation masks to identify contact regions and guide keypoint sampling, we observe that the most informative keypoints are empirically characterized by large motion magnitudes. Thus, we adaptively select the top-$K$ most active keypoints based on their motion magnitudes $\mathcal{M}_i = \sum_{t=1}^{T-1} ||\mathbf{p}_i^{t+1} - \mathbf{p}_i^t||_2, \forall i \in 1, ..., N_0$, producing the sampled trajectory set $\mathcal{T}_{sample} = \{(\mathbf{p}_i^1, ..., \mathbf{p}_i^T)\}_{i=1}^K$.

To enhance the keypoint dynamic learning, we encourage temporal consistency between the visual tokens of sampled keypoints by aligning all frames to the initial frame ($t = 1$) through the following loss:

$$\mathcal{L}_{\text{Keypoint}} = \frac{1}{(T-1)K} \sum_{i=1}^K \sum_{t=2}^T ||\hat{\mathbf{s}}_t(\mathbf{p}_i^t) - \hat{\mathbf{s}}_1(\mathbf{p}_i^1)||_2^2, \tag{5}$$

where $\hat{\mathbf{s}}_t(\mathbf{p}_i^t) \in \mathbb{R}^D$ denotes the $i$-th keypoint-located predicted token at frame $t$.

Furthermore, we observe that these dynamically active keypoint regions often exhibit higher token errors due to their complex motion patterns. To address this, we propose a keypoint-guided attention mechanism that adaptively enhances token learning in regions intersected by keypoint trajectories. Specifically, we compute a spatiotemporal attention map $\mathbf{A} \in \mathbb{R}^{T \times H' \times W'}$, with each element defined as:

$$\mathbf{A}_{t,x,y} = \begin{cases} \gamma & \text{if } (t, x, y) \in \mathcal{T}_{sample}, \\ 1 & \text{otherwise}, \end{cases} \tag{6}$$

where $\gamma$ is a hyperparameter controlling the importance weight. The attention-augmented training objective is formulated as:

$$\mathcal{L}_{\text{Attention}} = -\sum_{t=1}^T \mathbf{A}_t \odot \mathbf{s}_t \log p(\hat{\mathbf{s}}_t). \tag{7}$$

**Physics-informed Joint Training Objectives.** By integrating the above designs, we train a unified physics-aware world model through multi-task learning, with the final objective formulated as:

$$\mathcal{L} = \mathcal{L}_{\text{RGB}} + \lambda_1 \mathcal{L}_{\text{Depth}} + \lambda_2 \mathcal{L}_{\text{Keypoint}} + \lambda_3 \mathcal{L}_{\text{Attention}}, \tag{8}$$

where $\lambda_1, \lambda_2, \lambda_3 \in \mathbb{R}^+$ are are tunable coefficients balancing the loss terms.

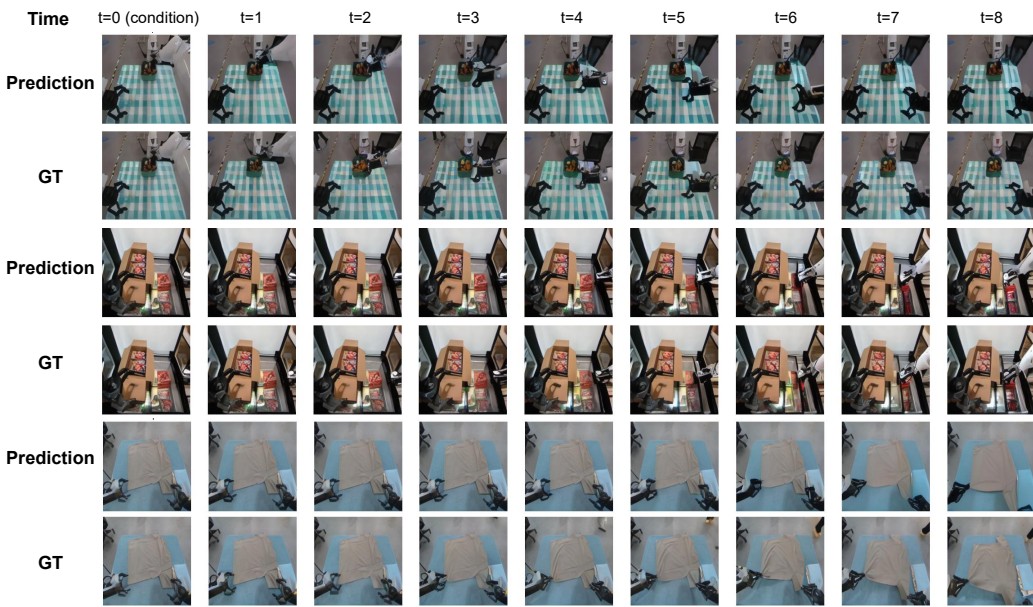

Figure 3: Qualitative results visualization of our model (only the subsequent 8 frames are shown). More results can be found in the appendix.

# 3 Experiments

In this section, we begin by detailing our experimental protocol (Section 3.1), including the dataset statistics, baseline information, and the implementation of our model. We then evaluate our model from three aspects: video quality evaluation (Section 3.2), robot policy learning with synthetic data (Section 3.3), and robotic policy evaluation (Section 3.4).

## 3.1 Experimental Settings

**Dataset Statistics.** In our experiment, we use 50,000 videos extracted from the AgiBotWorld-Beta dataset [40], covering 147 tasks and 72 skills. We concatenate the end position, end orientation, and effector position of the embodiment as the action sequence. Our dataset comprises approximately 6.5M training clips and 1.2K test clips.

**Baselines.** We compare our model with four advanced baselines, including both embodied world models (IRASim [14] and iVideoGPT [13]) and general world models (Genie [41] and CogVideoX [42]). Due to unavailable training codes in some recent works [29, 30], these methods are excluded from direct comparison. Details of baselines are presented in the appendix.

**Implementation Details.** We preprocess videos by extracting 16-frame clips sampled at 2Hz, yielding approximately 6.5 million training clips. The model is trained for 5 epochs using the following hyperparameters: $\lambda_1 = 1$, $\lambda_2 = 0.01$, $\lambda_3 = 1$, and $\gamma = 5$. Training completes in approximately 24 hours on a cluster of 32 NVIDIA A800-SXM4-80GB GPUs. During inference, we use the first frame as a conditional input to autoregressively predict the subsequent 15 frames.

## 3.2 Video Quality Evaluation

We evaluate video generation quality through three key dimensions: appearance fidelity, geometric consistency and action controllability. The details of the six used metrics are as follows:

- **PSNR**: It measures pixel-level reconstruction accuracy between generated and ground-truth frames.
- **LPIPS**: It assesses perceptual quality using visual feature similarity.
- **AbsRel**: It computes relative depth estimation errors.
- $\delta_1 / \delta_2$: They evaluate depth prediction accuracy at different precision levels.

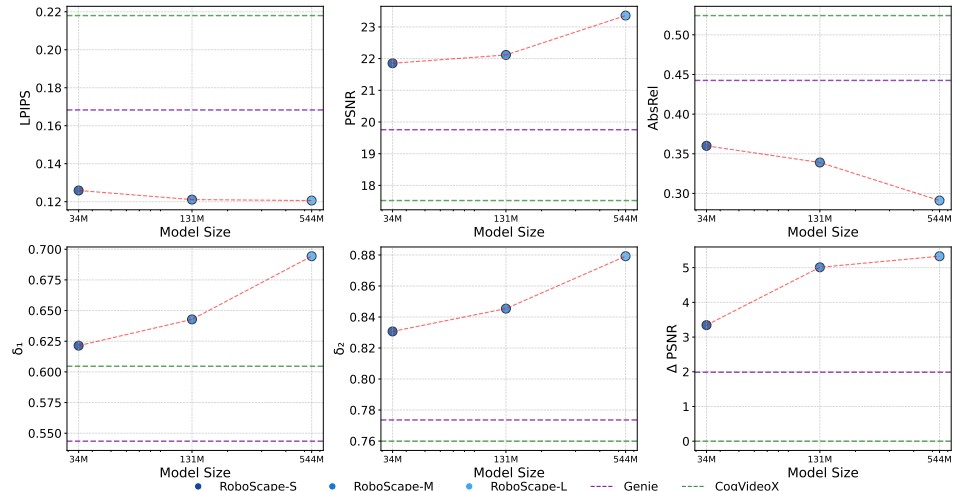

Figure 4: Model scaling law of RoboScape.

Table 1: Quantitative comparison of our model and baselines with 5 independent runs.

| Method | Appearance Fidelity | | Geometric Consistency | | | Action Controllability |
|---|---|---|---|---|---|---|
| | LPIPS ($\downarrow$) | PSNR ($\uparrow$) | AbsRel ($\downarrow$) | $\delta_1$ ($\uparrow$) | $\delta_2$ ($\uparrow$) | $\Delta$PSNR ($\uparrow$) |
| IRASim | 0.663±0.005 | 11.382±0.122 | 0.641±0.030 | 0.508±0.014 | 0.701±0.026 | 0.160±0.122 |
| iVideoGPT | 0.501±0.003 | 16.234±0.038 | 0.749±0.018 | 0.364±0.012 | 0.586±0.010 | 0.148±0.067 |
| Genie | 0.165±0.002 | 19.875±0.064 | 0.446±0.014 | 0.545±0.011 | 0.778±0.016 | 1.868±0.090 |
| CogVideoX | 0.202±0.008 | 17.957±0.153 | 0.508±0.026 | 0.594±0.017 | 0.739±0.016 | — |
| **RoboScape** | **0.128±0.002** | **21.730±0.120** | **0.378±0.026** | **0.608±0.011** | **0.814±0.013** | **3.442±0.139** |

- **ΔPSNR**: It quantifies output sensitivity to action condition, with higher values indicating better action control ability.

We present some generation results in Figure 3, where we predict future frames conditioned on an initial frame and robot action commands (we visualize 8 frames while the model supports long-horizon rollouts). The visualizations demonstrate that our model effectively simulates realistic robot manipulation scenarios, with generated sequences showing strong similarity to ground truth observations. Notably, our approach successfully handles deformable object interactions, as evidenced by the cloth-dragging sequence where the generated deformations accurately follow physical laws and capture material properties.

As shown in Table 1, we conduct comprehensive comparisons with four advanced baselines: two embodied world models (IRASim and iVideoGPT) and two general world models (Genie and CogVideoX). Our model consistently outperforms all baselines across six evaluation metrics, demonstrating its superior capability in video prediction for robotic scenarios. Detailed analysis reveals that while CogVideoX can generate high-quality videos, its inability to follow action commands leads to substantial deviations in future frames. The two embodied world models are not good at motion learning when conducting long-term generation, thus receiving poor metrics. Our model's novel integration of keypoint dynamics learning effectively addresses these limitations, simultaneously achieving high-fidelity visual generation and superior action controllability.

We further conduct ablation studies to demonstrate the complementary benefits of our two core components: temporal depth learning and keypoint dynamics learning. The results are shown in Table 2. The quantitative results reveal that both components contribute significantly to overall performance; removing either one leads to measurable degradation across different metrics. The depth learning primarily preserves geometric consistency of moving objects, and the keypoint learning proves essential for maintaining both visual fidelity and action controllability. We provide a case study in Figure 6. It can be seen that the missing of temporal depth learning will lead to geometric distortions in moving objects, while the absence of key-point dynamics learning results in unreal motion patterns. These findings collectively validate the necessity of our key designs.

Table 2: Ablation study of our key designs of physics prior injection with 5 independent runs.

| Method | LPIPS ($\downarrow$) | PSNR ($\uparrow$) | AbsRel ($\downarrow$) | $\delta_1$ ($\uparrow$) | $\delta_2$ ($\uparrow$) | $\Delta$PSNR ($\uparrow$) |
|---|---|---|---|---|---|---|
| whole model | 0.128±0.002 | 21.730±0.120 | 0.378±0.026 | 0.608±0.011 | 0.814±0.013 | 3.442±0.139 |
| w/o depth | 0.126±0.001 | 21.885±0.046 | 0.408±0.010 | 0.560±0.012 | 0.789±0.022 | 3.514±0.023 |
| w/o keypoint | 0.128±0.001 | 21.634±0.043 | 0.346±0.012 | 0.637±0.012 | 0.848±0.012 | 2.953±0.036 |
| w/o depth & keypoint | 0.130±0.001 | 21.477±0.029 | 0.371±0.012 | 0.598±0.018 | 0.800±0.009 | 1.945±0.054 |

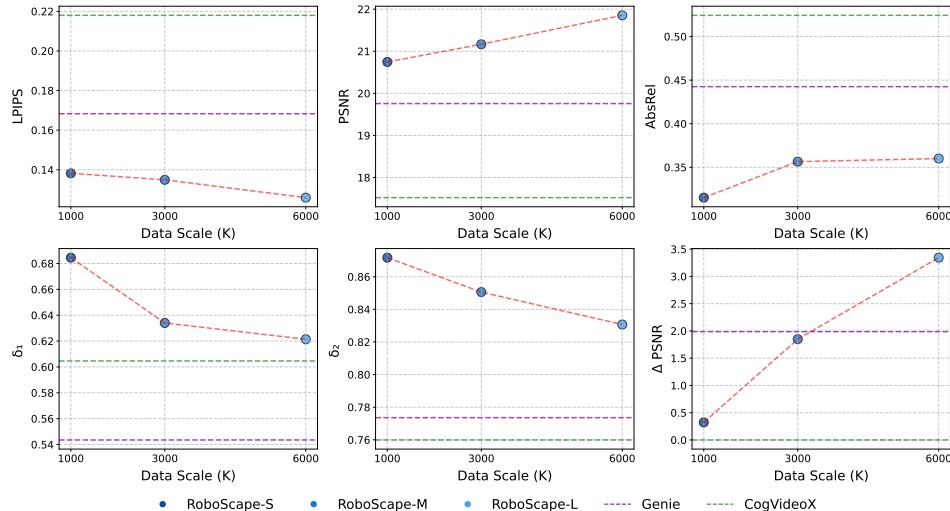

Figure 5: Data scaling law of RoboScape-S.

We also investigate the scaling behavior of RoboScape in terms of both model and data scales. As shown in Figure 4, we evaluate three model variants—RoboScape-S (34M), RoboScape-M (131M), and RoboScape-L (544M)—and observe a clear scaling law: all six evaluation metrics improve significantly as model capacity increases. In addition, we study the impact of data scale by training RoboScape-S on 1,000K, 3,000K, and 6,000K clips (Figure 5). While increasing data size consistently enhances visual quality and action controllability, geometric accuracy exhibits marginal improvement or even slight degradation. We find that this is because smaller datasets encourage overfitting to the final frame of conditional inputs, artificially inflating geometric metrics without generating meaningful temporal dynamics. Despite this, the overall trend confirms that more training data leads to better model performance.

### 3.3 Robotic Policy Learning with Synthetic Data

We validate our world model's utility by generating synthetic robotic video data for downstream policy learning based on Diffusion Policy (DP) [31] and $\pi 0$ [32]. The actions and initial observations for our synthetic data are directly drawn from the raw RoboMimic and LIBERO datasets. Through controlled experiments with progressively adding synthetic data, we systematically measure the impact of generated data on policy learning performance. The results are shown in Table 3. "Real data" mentioned consists of original videos with their corresponding action annotations from the raw dataset. For "Synthetic Data", we use our world model to generate the videos based on actions and initial observations.

In the experiments on the Robomimic Lift task [43], DP trained for 10k steps with only generated data achieved nearly the same performance as DP trained with real data. Notably, the policy success rate exhibited consistent improvement with increasing synthetic training data, highlighting the effectiveness of our model. We further validated our approach using the $\pi_0$ [32] model on the challenging LIBERO [44] task suite. These tasks present three key challenges beyond the Robomimic Lift environment: (1) complex multi-object manipulation requirements, (2) cluttered scene configurations, and (3) extended action sequence horizons. Therefore, we employ a small amount of real data (200 trajectories) as a training warm-up. Remarkably, when training $\pi_0$ policies with increasing generated data, the model performance achieves gradual improvement. These results

Table 3: Results of policy learning with DP on Robomimic task and $\pi 0$ on LIBERO tasks.

| DP on Robomimic tasks | | $\pi 0$ on LIBERO tasks | | | | | |
|---|---|---|---|---|---|---|---|
| # Synthetic Data | Success Rate | #Synthetic Data | Spatial | Object | Goal | 10 | Average |
| 50 | 40% | 200 | 77.6% | 81.8% | 71.0% | 36.0% | 66.6% |
| 100 | 77% | 400 | 79.4% | 85.2% | 74.6% | 46.2% | 71.4% |
| 150 | 84% | 600 | 81.6% | 86.0% | 78.0% | 51.8% | 74.4% |
| 200 | 91% | 800 | 84.6% | 89.0% | 82.8% | 60.0% | 79.1% |
| Real (200) | 92% | Real (200) | 77.2% | 79.8% | 68.8% | 34.8% | 65.2% |

Figure 6: Effect of the physics knowledge learning. Omission of temporal depth learning leads to geometric distortions in moving objects, while the absence of key-point dynamics learning results in unreal motion patterns.

demonstrate our model's capability to generate physically plausible trajectories even for demanding, long-horizon manipulation scenarios.

## 3.4 Robotic Policy Evaluation

In this section, we investigate whether our world model can act as a policy evaluator for different robotic policies. In policy evaluation, the world model acts as an environment that receives policy-generated action sequences and predicts subsequent observations in a rollout manner. Policy quality is then assessed by checking success rates in the predicted videos. Here we compare IRASim, iVideoGPT, and our model as the policy evaluator and use Diffusion Policy [31] as the policy model. Specifically, we train the policy on the Robomimic Lift task [43] using 200 trajectories and save the policy every 250 epochs until it is fully converged. Then we post-train the world model and evaluate the policy in both the ground-truth simulator and the world model by 100 runs. The success signal of each run can be directly given by the simulator, while it requires manual judgment when the policy interacts with the world model. The generated videos from all models were presented to participants in a randomized, blind order, with no model identifiers displayed. Participants were instructed to judge whether the task depicted in each video was successfully completed. This setup ensures objectivity and impartiality. Afterwards, we calculate the Pearson correlation and $R^2$ between different world models and the ground-truth simulator. The results in Figure 7 show that the Pearson correlation of our model is **0.953**, while the correlation of other models is rather low, indicating that our world model can be utilized as a better policy evaluator.

## 4 Related Work

### 4.1 World Model

World models learn representations of environmental states through neural networks, enabling the prediction of future states based on current observations and actions [45]. Recent advances in world models primarily leverage video generation techniques, with applications spanning three key domains including autonomous driving [46, 47, 48, 49, 50, 51, 52], embodied intelligence [53, 14, 15], and

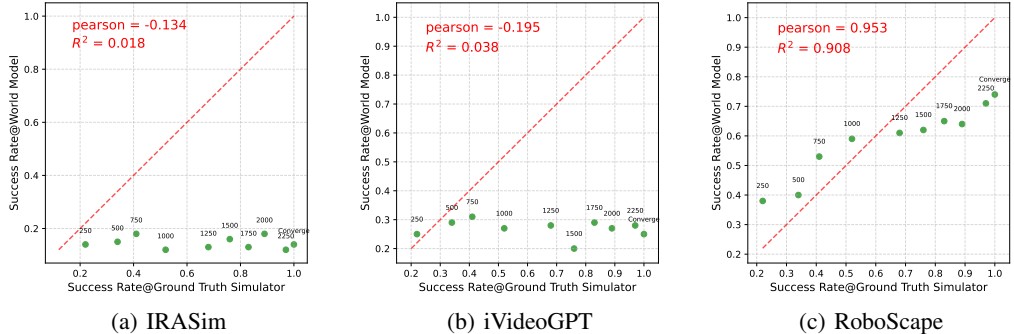

Figure 7: Correlation between the success rate of different world models and the ground-truth simulator. Each point represents a policy, and the trained epochs are shown above the point.

gaming [54, 55, 56, 48]. The dominant modeling approaches fall into two main categories: diffusion models and autoregressive models. Diffusion models, such as DiT [57], generate sequences through a gradual denoising process and are well-suited for producing diverse and short-term consistent visual content. Autoregressive models, such as Genie[41], reconstruct sequences via masking mechanisms and demonstrate superior efficiency and controllability. Compared to diffusion-based approaches, autoregressive methods offer advantages in inference speed and training stability. Our work utilizes masked autoregressive models with physical information injection, aiming to build efficient and interactive world models for embodied intelligence.

### 4.2 Physics-aware Generative Model

Recent advances in video generation have increasingly focused on improving the modeling of physical properties [17]. Current methods in this area can be roughly divided into explicit and implicit physical modeling. Explicit methods incorporate physical information by learning explicit textures and material representations [28, 30]. In contrast, implicit methods mainly embed physical knowledge into models via training loss terms [20], or by using generative models to jointly generate RGB videos and other physical representations [29, 58]. These approaches aim to enhance physical understanding through data-driven approaches rather than predefined physical rules. Currently, there's much room for existing embodied world models to enhance the integration of physical knowledge into video generation. To advance this field, we introduce a physics-informed embodied world model that jointly learns RGB video generation, temporal depth prediction, and keypoint dynamics within a unified framework, achieving both high visual fidelity and physical plausibility.

## 5 Conclusion and Future Work

In this work, we propose RoboScape, a physics-informed embodied world model that efficiently integrates physical knowledge into video generation through a physics-inspired multi-task joint training framework, eliminating the need for cascaded external models such as physics engines. By incorporating temporal depth prediction, our model learns the 3D geometric structure of scenes, while dynamic keypoint learning enables implicit modeling of object deformation and motion patterns. Extensive evaluations demonstrate that our approach outperforms baseline methods in video generation quality, synthetic data utility for downstream robotic manipulation policy training, and effectiveness as a policy evaluator. In the future, we plan to combine the generative world model with real-world robots to test performance further.

## Acknowledgments and Disclosure of Funding

This work was supported in part by the National Key Research and Development Program of China (No.2024YFC3307603).

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

## A    Broader Impacts

Our physics-informed world model offers positive impacts by advancing robotic learning: it generates high-fidelity synthetic data with inherent physical plausibility, which drastically reduces reliance on costly real-world data collection and improves sim-to-real transfer for applications in healthcare, disaster response, and industry. While this computational efficiency lowers research barriers, the high fidelity necessitates ethical consideration. Specifically, the generated content could be misused to create deepfakes or misrepresent safe procedures. To mitigate these risks, future work should focus on training fake detection models to identify synthetic content and incorporating anomaly detection mechanisms to flag or prevent the generation of videos depicting unsafe or non-standard robot behaviors, ensuring responsible deployment in safety-critical domains.

## B    Limitations

While our physics-informed world model achieves significant gains in physical consistency, we acknowledge the following shortcomings and inherent assumptions: (1) Our current physical constraints focus primarily on geometric (depth) and kinematic (keypoint dynamics) properties. This leaves out broader critical information such as dynamic force interactions. Furthermore, our model relies on the empirical assumption that keypoint dynamics implicitly capture material properties, which lacks a theoretical guarantee. (2) Our model's inference is currently limited to a 48-frame rollout. This computational constraint makes it challenging to simulate and evaluate very long-horizon tasks (e.g., complete cloth folding), where we have observed that cumulative error can lead to physically implausible outcomes. (3) Generalization to more diverse embodiments, such as quadrupeds or humanoids, is currently limited by the lack of readily available, high-quality, multi-domain datasets. Our current validation is primarily confined to single- and dual-arm manipulation settings. Future work will be dedicated to overcoming these limitations by incorporating richer physical laws, exploring more efficient hybrid architectures, and expanding our training to more diverse embodiment data.

## C    Baseline Details

We provide details of the compared baselines as follows:

- **IRASim**: A DiT-based robotic video generation model, capable of generating videos conditioned on robot actions and trajectories.

- **iVideoGPT**: An auto-regressive interactive world model that takes the current video frame observation and action as input to predict the next frame while simultaneously estimating the reward signal for robotic operations.

- **Genie**: A foundation world model trained through unsupervised learning on massive video data. We implement it with a reproduced open-source repository [*].

- **CogVideoX**: An advanced DiT-based text-to-video generation framework, with superior performance in prompt-driven video generation.

## D    Supplemented Evaluation toward Physics Correctness

In this section, we provide a comparison regarding the physical correctness of our model and baselines. While existing physics benchmarks for video generation are primarily designed for general text-to-video models and aren't directly applicable to our embodied world model (which is driven by robotic actions), we've identified that key metrics from these benchmarks can be utilized for our evaluation. Specifically, we utilize four metrics from the Physics-IQ benchmark [59] that are particularly effective in assessing the physical realism of motion plausibility and object deformation, detailed as follows:

- **Spatial IoU**: Measures "Whether the location where an action happens is correct".

- **Spatiotemporal IoU**: Measures "Where does action happen and whether it occurs at the right time".

---

[*]https://github.com/1x-technologies/1xgpt

Table 4: Comparison of world models on physics correctness.

| Model | Spatial IoU (↑) | Spatiotemporal IoU (↑) | Weighted Spatial IoU (↑) | MSE (↓) |
|---|---|---|---|---|
| IRASim | 0.2431 | 0.1255 | 0.2081 | 0.0655 |
| Genie | 0.6429 | 0.3420 | 0.6082 | 0.0397 |
| CogVideoX | 0.6523 | 0.2058 | 0.4675 | 0.0943 |
| **RoboScape** | **0.7573** | **0.4454** | **0.7023** | **0.0184** |

Table 5: Policy success rate using generated data from different world models on LIBERO benchmark.

| Model | LIBERO-Spatial | LIBERO-Object | LIBERO-Goal | LIBERO-10 | Average |
|---|---|---|---|---|---|
| Only real data | 77.2% | 79.8% | 68.8% | 34.8% | 65.2% |
| IRASim | 72.8% | 77.4% | 75.2% | 34.0% | 64.8% |
| CogVideoX | 81.0% | 79.6% | 74.4% | 44.2% | 69.8% |
| RoboScape | **84.6%** | **89.0%** | **82.8%** | **60.0%** | **79.1%** |

- **Weighted Spatial IoU**: Measures "Where and how much action happens".

- **MSE**: Measures "How objects look and interact" at pixel-level.

  The results shown in Table 4 demonstrate our model's superior performance in terms of physical correctness.

## E    Supplemented Results of Policy Learning

We compare training the pi0 model using data synthesized by baseline world models and our model. For fair comparisons, both CogVideoX and IraSim were fine-tuned on the LIBERO dataset, with each model generating 800 synthetic data points. Table 5 presents the experimental results across various task subsets in the LIBERO environment, indicating the superiority of our model in generating synthetic data for VLA training compared to baselines.

## F    More Visualization Results

### F.1    Video Generation Results

We provide more visualization results of generated videos using our model, as illustrated in Figure 8.

### F.2    Robotic Policy Learning

We provide some visualization results of generated data on Robomimic and LIBERO using our model, which are shown in Figure 9 and Figure 10.

### F.3    Robotic Policy Evaluation (add visualization results of our model and baselines

In this part, we provide visualization results of RoboScape and other baselines in policy evaluation. The failure cases are presented in Figure 11 while the successful cases are shown in Figure 12.

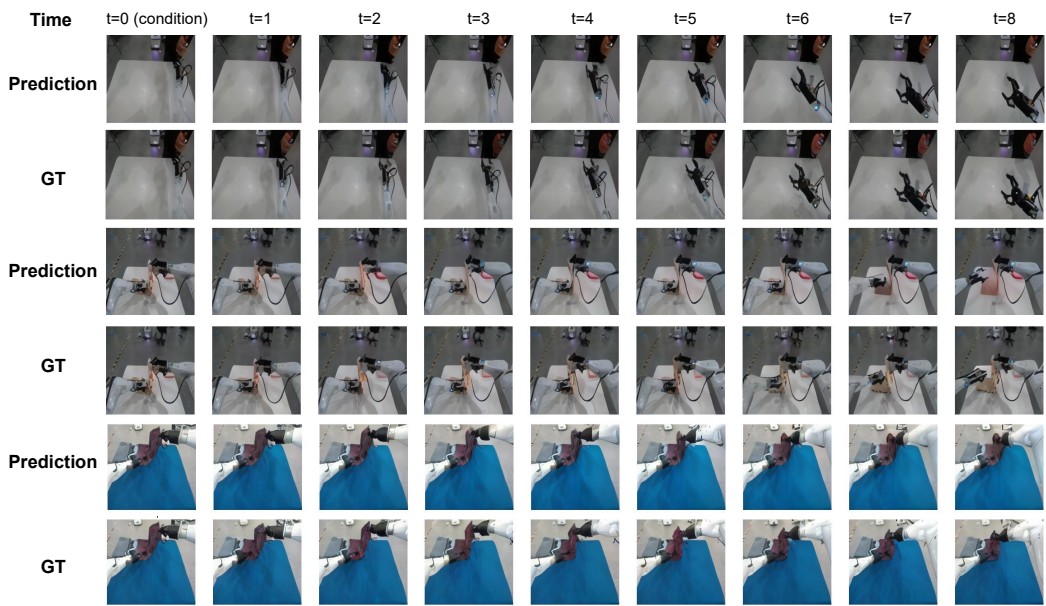

Figure 8: Supplemented visualization results from our model (only the subsequent 8 frames are shown).

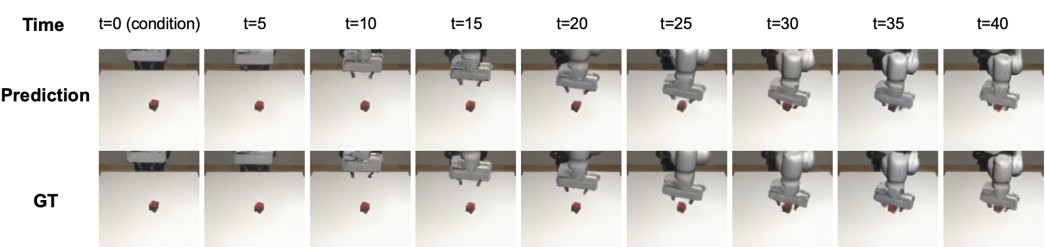

Figure 9: Supplemented visualization results on Robomimic (displaying every 5th frame; 8 frames shown from t=0 to t=40).

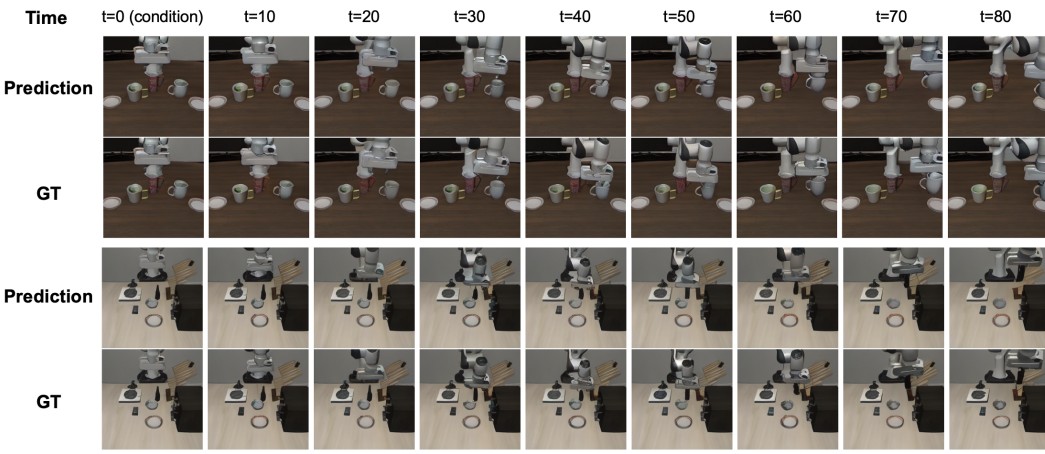

Figure 10: Supplemented visualization results on LIBERO (displaying every 10th frame; 8 frames shown from t=0 to t=80).

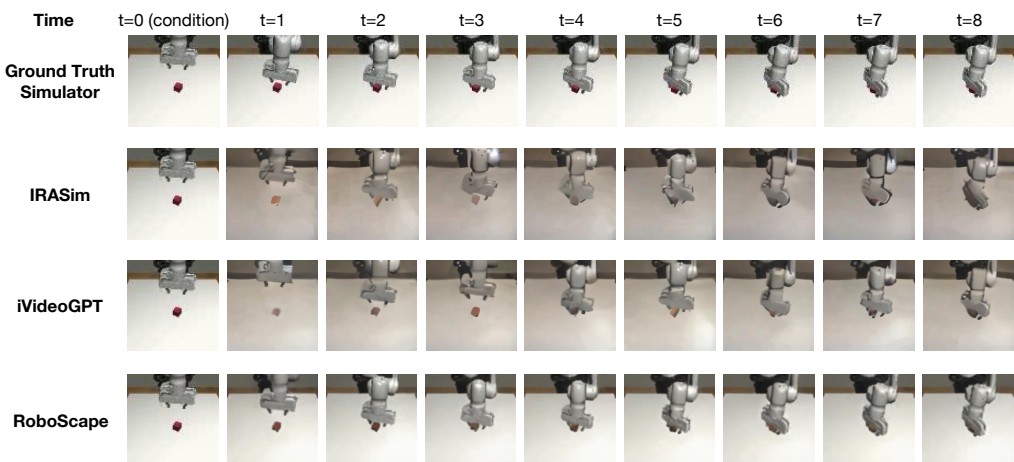

Figure 11: Supplemented visualization results of failure cases in policy evaluation.

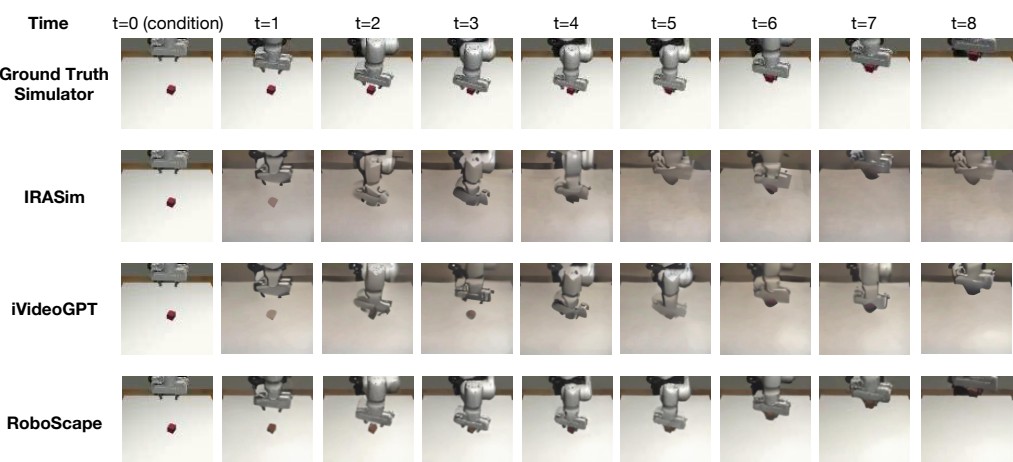

Figure 12: Supplemented visualization results of successful cases in policy evaluation.

