# OpenReview forum: "RoboScape: Physics-informed Embodied World Model"
_NeurIPS.cc/2025/Conference — NeurIPS 2025 spotlight_

### Official Review · Reviewer_isFR · 2025-07-02

**Clarity:** 4
**Significance:** 3
**Originality:** 3
**Rating:** 5
**Confidence:** 3

**Summary:**

This paper presents a world model for robot learning. The proposed method, RoboScape, learns to predict video and depth images that are physically consistent. Physical knowledge is transferred through a proxy task of keypoint tracking, where latent features of keypoints are enforced to be consistent across time. Experiments show that the proposed method can be used to generate synthetic data for robot training or evaluate robot policies.

**Questions:**

Please refer to the "weaknesses" section.

**Ethical Concerns:**

["NO or VERY MINOR ethics concerns only"]

**Final Justification:**

The rebuttal has allayed all my concerns, and I especially appreciate the new sets of experiments conducted using $\pi0$. Thus I am keeping my initial rating of accept.

**Limitations:**

The authors have stated their limitations in the "Conclusion" section.

**Paper Formatting Concerns:**

No concerns exist.

**Quality:**

4

**Strengths And Weaknesses:**

[Strengths]
1. The writing is clear, and the paper is well-motivated. World models that are capable of generating physically consistent videos have large potential applications in robotics, and the paper suggests a solution to the problem.

2. Performance-wise, the paper outperforms existing world models in terms of video prediction performance.

3. I find the experiment in Section 3.4 to be particularly interesting, suggesting the possibility of RoboScape to be used as a policy evaluator.

[Weaknesses]
1. The paper currently only uses the world model for generating synthetic data. I am curious how the authors could formulate the world model directly for planning, similar to the recently proposed V-JEPA2 work [1].

2. The performance against the baselines are all measured in terms of video prediction quality. Would it be possible to perform comparisons in the actual task domain, i.e., generate synthetic data using each model and compare $\pi 0$ model's performance? This would better highlight the effectiveness of the proposed method for robot data generation.

3. Have the authors also tried giving "dense supervisions" for world model training, namely via optical flow? I am curious how the performance would differ if the authors used the loss function from Equation 5 with a *shorter* timespan but with *denser* pixels.

[Related Work]
1. Assran et al., V-JEPA 2: Self-Supervised Video Models Enable Understanding, Prediction and Planning, arXiv 2025.

---

> ### Author Rebuttal · Authors · 2025-07-31
>
> **Q1: The paper currently only uses the world model for generating synthetic data. I am curious how the authors could formulate the world model directly for planning, similar to the recently proposed V-JEPA2 work.**
>
> **Response:**
>
> Thank you for this insightful question. Our model can also be directly used for action planning, much like the approach in V-JEPA 2. In fact, both video-generative world models and JEPAs are fundamentally modeling the dynamics of state transitions, with the key difference being whether the world state is represented in the pixel space or a latent space.
>
> Therefore, our model can be formulated for action planning using a model predictive control (MPC) approach. Specifically, we can sample multiple potential action trajectories and use our world model to predict the future frames for each trajectory. By using the similarity between the generated final frame and a target goal image as an energy function, we can select the action sequence that minimizes this energy as the result. We will adopt this evaluation as another evaluation dimension of our world model.
>
> **Q2: The performance against the baselines are all measured in terms of video prediction quality. Would it be possible to perform comparisons in the actual task domain, i.e., generate synthetic data using each model and compare π0 model's performance? This would better highlight the effectiveness of the proposed method for robot data generation.**
>
> **Response:** We appreciate this valuable suggestion, which indeed offers a more direct assessment of our method's effectiveness in robot data generation. We've conducted additional experiments comparing the π0 model's performance when trained on synthetic data generated by different world models. Following the setup in Table 3, we compared training the π0 model using data synthesized by **CogVideoX** and **IraSim** (experiments for IraSim are ongoing and will be updated soon). For fair comparisons, both CogVideoX and IraSim were fine-tuned on the **LIBERO dataset**, with each model generating 800 synthetic data points. The table below presents the experimental results across various task subsets in the LIBERO environment, indicating the superiority of our model in generating synthetic data for VLA training compared to baselines.
>
> | **Model**      | LIBERO-Spatial | LIBERO-Object | LIBERO-Goal | LIBERO-10 | Average |
> | -------------- | -------------- | ------------- | ----------- | --------- | ------- |
> | Only real data | 77.2%          | 79.8%         | 68.8%       | 34.8%     | 65.2%   |
> | CogVideoX      | 81.0%          | 79.6%         | 74.4%       | 44.2%     | 69.8%   |
> | RoboScape      | 84.6%          | 89.0%         | 82.8%       | 60.0%     | 79.1%   |
>
>
>
> **Q3: Have the authors also tried giving "dense supervisions" for world model training, namely via optical flow? I am curious how the performance would differ if the authors used the loss function from Equation 5 with a *shorter* timespan but with *denser* pixels.**
>
> **Response:**
>
> Thanks for this insightful question. Our motivation for learning keypoint consistency (Equation 5) is its **inherent inter-frame consistency for the same keypoint in the RGB space**. This provides a natural physical constraint that significantly enhances the model's **understanding of object deformation and motion dynamics**, implicitly aiding in capturing material properties like rigidity and softness. What's more, the number of keypoints is a tunable hyper-parameter, also allowing for denser supervision as you mentioned, though this might impact training speed.
>
> Regarding optical flow, as you correctly noted, it provides a dense signal representing pixel motion vectors between consecutive frames. However, **optical flow feature maps between different frames are not necessarily continuous or strictly inter-frame consistent**. This inherent lack of strict consistency makes it an unsuitable constraint for direct application using Equation 5. Our chosen key-point supervision provides the necessary inter-frame consistency to effectively guide the model's learning of object dynamics and properties.

---

> > ### Comment · Reviewer_isFR · 2025-08-05
> >
> > I thank the authors for the rebuttal. The rebuttal has clarified all my concerns, and I greatly appreciate the new sets of experiments done using the $\pi0$ model. Thus I'm keeping my original score of accept.

---

> > > ### Author Response · Authors · 2025-08-05
> > >
> > > Dear Reviewer isFR,
> > >
> > > Thanks for your valuable time and positive feedback. We are delighted that our responses have effectively clarified all your concerns. Your suggestions for our work are greatly valued.

---

> ### Author Response · Authors · 2025-08-01
> **Updated experiment results in response to Q2**
>
> The updated experimental results of pi0 on various LIBERO task subsets are as follows:
> | **Model**      | LIBERO-Spatial | LIBERO-Object | LIBERO-Goal | LIBERO-10 | Average |
> | -------------- | -------------- | ------------- | ----------- | --------- | ------- |
> | Only real data | 77.2%          | 79.8%         | 68.8%       | 34.8%     | 65.2%   |
> | IRASim         | 72.8%          | 77.4%         | 75.2%       | 34.0%     | 64.8%   |
> | CogVideoX      | 81.0%          | 79.6%         | 74.4%       | 44.2%     | 69.8%   |
> | RoboScape      | 84.6%          | 89.0%         | 82.8%       | 60.0%     | 79.1%   |

---

### Official Review · Reviewer_NSMY · 2025-07-03

**Clarity:** 2
**Significance:** 2
**Originality:** 2
**Rating:** 4
**Confidence:** 4

**Summary:**

This paper proposes integrating physics into embodied world models to generate physically accurate robotic videos. It utilizes two main components: temporal depth prediction to improve 3D consistency and keypoint tracking to learn dynamics. The resulting model demonstrates enhanced performance in downstream robotic policy training.

**Questions:**

- Can the authors elaborate on the scope of the dataset varieties in terms of environment and robot settings? For example, what types of robots, objects, and scene settings are included, and how are they distributed? What robotic tasks are modeled? It helps to assess the breadth of this contribution knowing how general the dataset is.

- The authors have specifically chosen two features in this work (depth and keypoint trajectory), both of which have been explored in previous studies to improve subject consistency and motion smoothness in generated video. How does this approach differ from previous work, perhaps in terms of model architecture or training strategies? Additionally, why were these features chosen for this study? Further insight into these aspects may help clarify the unique contribution of this work.

- Can the authors provide more details on the model architecture design?

**Ethical Concerns:**

["NO or VERY MINOR ethics concerns only"]

**Final Justification:**

The authors have adequately addressed most of my concerns. Given the unavailability of the visual content, I would ask them to include the additional results as discussed in their rebuttal to Q1 and Q2.

**Limitations:**

See "weaknesses" and "questions" above.

**Paper Formatting Concerns:**

None.

**Quality:**

2

**Strengths And Weaknesses:**

Strengths:

The proposed method outperforms various baselines in terms of appearance fidelity and geometry consistency.

Weaknesses:

- The paper claims that existing embodied/general world models often exhibit artifacts like morphing or discontinuous motion. However, it lacks visual illustrations of these failure cases. To better showcase the robustness of the proposed method, a more extensive qualitative comparison (e.g., generation results under the same prompts) is needed, in addition to the current quantitative comparisons.

- Also, the absence of accompanying videos for visual illustration. This makes it difficult to evaluate the visual quality, particularly the smoothness of motion.

- Additionally, there’s concern regarding the variety of the dataset and trained world model. The paper only covers certain robot arm settings, leaving out many other embodied models such as humanoid robots, UAVs, quadrupeds, and soft robots. This limitation could stem from the scalability of the data, which might be a potential barrier.

---

> ### Author Rebuttal · Authors · 2025-07-31
>
> **Q1: The paper claims that existing embodied/general world models often exhibit artifacts like morphing or discontinuous motion. However, it lacks visual illustrations of these failure cases. To better showcase the robustness of the proposed method, a more extensive qualitative comparison (e.g., generation results under the same prompts) is needed, in addition to the current quantitative comparisons.**
>
> **Response:**
> We appreciate this valuable feedback. We acknowledge that a visual demonstration would enhance our claims. While we cannot provide images or videos during this rebuttal phase due to formatting limitations, we assure that we will add it in the revised paper.
>
>
>
> **Q2: Also, the absence of accompanying videos for visual illustration. This makes it difficult to evaluate the visual quality, particularly the smoothness of motion.**
>
> **Response:** Thank you for the suggestion. Similar to the response to Q1, while we cannot provide videos during the rebuttal phase due to formatting limitations, we will establish a project page for the paper to include more videos to further demonstrate the visual quality.
>
>
>
> **Q3: Additionally, there’s concern regarding the variety of the dataset and trained world model. The paper only covers certain robot arm settings, leaving out many other embodied models such as humanoid robots, UAVs, quadrupeds, and soft robots. This limitation could stem from the scalability of the data, which might be a potential barrier.**
>
> **Response:**
> We appreciate this point. Our current data already includes a range of manipulation scenarios with both single-arm and dual-arm gripper-based robots, which have varying degrees of freedom. By adapting the action encoder, our framework can be easily generalized to work with data from different robots. Just as you mentioned, currently high-quality data and testbeds for other types of robots like humanoid robots, UAVs, quadrupeds, and soft robots are limited. As these resources become more widely available, we plan to expand our training data and validate our model on these diverse robot morphologies.
>
>
>
> **Q4: Can the authors elaborate on the scope of the dataset varieties in terms of environment and robot settings? For example, what types of robots, objects, and scene settings are included, and how are they distributed? What robotic tasks are modeled? It helps to assess the breadth of this contribution knowing how general the dataset is.**
>
> **Response:**
> We utilized the Agibot-World dataset [1], a recently released large-scale robotic learning dataset collected with humanoid robots. Specifically, we focused on data from dual-arm end-effector type robots, excluding those with dexterous hands. The data we used included 147 tasks (e.g. closing the refrigerator door, placing objects, etc.), and 72 skills (e.g. pick, push, etc.). The dataset covers 5 scenes including home (40%), dining (20%), industry (20%), supermarkets (10%) and office (10%) and each scene consists of multiple environments.
>
> [1] https://huggingface.co/agibot-world
>
>
>
> **Q5: The authors have specifically chosen two features in this work (depth and keypoint trajectory), both of which have been explored in previous studies to improve subject consistency and motion smoothness in generated video. How does this approach differ from previous work, perhaps in terms of model architecture or training strategies? Additionally, why were these features chosen for this study? Further insight into these aspects may help clarify the unique contribution of this work.**
>
> **Response:**
> Existing methods using extra depth or motion information for video generation typically fall into two categories:
>
> - **As extra channels:** Some approaches, like those in [1] and [2], treat depth or motion as additional channels alongside RGB, using a diffusion model to denoise them jointly. The key limitation here is the lack of explicit supervision, leading to a weak coupling between these channels.
> - **As external conditions:** Other methods, such as those in [3] and [4], use depth or motion as external conditions to guide video generation. This approach restricts flexibility during inference, as it requires pre-annotated depth or motion information.
>
> [1] Team, Aether, et al. "Aether: Geometric-aware unified world modeling." *arXiv preprint arXiv:2503.18945* (2025).
>
> [2] Xi, Dianbing, et al. "Omnivdiff: Omni controllable video diffusion for generation and understanding." *arXiv preprint arXiv:2504.10825* (2025).
>
> [3] Esser, Patrick, et al. "Structure and content-guided video synthesis with diffusion models." ICCV 2023.
>
> [4] Wang, Xiang, et al. "Videocomposer: Compositional video synthesis with motion controllability." NeurIPS 2023.
>
> **Our differences with prior works:**
>
> - **Model Architecture:** Unlike the prevalent use of diffusion models, we employ an autoregressive model, which is more suitable for interactive world modeling. Our architecture includes two parallel branches for RGB and depth prediction. Crucially, the intermediate features from the depth prediction branch are explicitly injected into the RGB prediction branch, providing direct guidance for video frame synthesis. Such dual branch-based world model architecture is a novel design.
> - **Training Strategy:** We employ a novel multi-task learning strategy that **jointly learns world model dynamics and physical constraints**. Our model is trained to predict both the next RGB frame and its corresponding depth map in an autoregressive manner, driven by robot actions. The predicted depth information then acts as an explicit constraint on the RGB prediction process. Additionally, we use the temporal consistency of keypoints as a physical prior to locally constrain the RGB frames, enhancing the plausibility of deformation and motion. This unified training paradigm is distinct from the conditional diffusion training commonly used in prior works.
>
> **Motivation for our feature choices:**
>
> We chose to focus on depth prediction and keypoint dynamic learning for two primary reasons:
>
> - **Addressing key robotic manipulation challenges:** General video generation models often struggle with two critical issues: unrealistic object deformation and motion, and inaccurate scene geometry. Both of these are fatal flaws when creating synthetic data for robot tasks. By incorporating depth and keypoint dynamics, we directly impose physical constraints that correct these deficiencies, leading to more physically plausible robotic manipulation videos.
>
> - **Practicality and generalizability:** These features are highly practical because they can be clearly and directly expressed through visual information. This allows for seamless integration into our video generation-based world model. Moreover, annotations for depth and keypoints are widely available and can be extracted using existing tools, which ensures our method's scalability and generalizability to various datasets.
>
>
>
> **Q6: Can the authors provide more details on the model architecture design?**
>
> **Response:**
> Our framework is composed of four main modules: an action encoder, a visual tokenizer, and two parallel prediction branches for RGB and depth images, detailed as follows:
>
> - **Visual Tokenizer:** We uniformly use MAGVIT-2 to compress both RGB frames and depth maps into discrete tokens. All subsequent video frame prediction is conducted within this token space.
> - **Action Encoder:** A simple linear layer is used to map the 16-dimensional robot action vector into a latent embedding space (with a default dimension of 256).
> - **RGB and Depth Prediction Branches:** Both branches consist of a stack of 32 spatial-temporal attention blocks. Each block is a sequence of spatial attention, temporal attention, and an MLP layer with residual connections. Both spatial and temporal attention layers have 8 attention heads, and the latent embedding dimension is set to 256. The RGB branch takes the fused action and current RGB features as input to predict the next RGB features. The depth branch does the same for depth. A key architectural detail is that intermediate features from the depth branch are explicitly injected into the corresponding layers of the RGB branch to provide direct geometric guidance and enhance the RGB feature representation.
>
> We are willing to provide further details on any specific part of the architecture that you are concerned about.

---

> > ### Comment · Reviewer_NSMY · 2025-08-05
> >
> > Thank you for the rebuttal. One major concern in my original response was the absence of a comparison regarding the physical correctness of the model against previous models (e.g., unrealistic object morphing or discontinuous motion). Given the unavailability of visual content, have the authors attempted any quantitative measurements of the physics aspects? Several benchmarks are available, for example:
> > [1] (PhyGenBench) Towards World Simulator: Crafting Physical Commonsense-Based Benchmark for Video Generation
> > [2] VideoPhy: Evaluating Physical Commonsense for Video Generation
> > [3] Physics-IQ: Benchmarking Physical Understanding in Generative Video Models
> >
> > I understand if this is undoable given the time constraints, but I believe it would strengthen the paper’s claim of a physics-informed world model.

---

> ### Author Response · Authors · 2025-08-06
>
> Dear Reviewer NSMY,
>
> Thanks for your valuable suggestion. We agree that a comparison regarding the physical correctness would strengthen the paper's claim of a physics-informed world model. While existing physics benchmarks for video generation are primarily designed for general text-to-video models and aren't directly applicable to our embodied world model (which is driven by robotic actions), we've identified that key metrics from these benchmarks can be utilized for our evaluation.
>
> Specifically, we've utilized four metrics from the **Physics-IQ benchmark** [1] that are particularly effective in assessing the physical realism of motion plausibility and object deformation,  detailed as follows:
>
> - **Spatial IoU:** Measures "Whether the location where an action happens is correct".
> - **Spatiotemporal IoU:** Measures "Where does action happen and whether it occurs at the right time".
> - **Weighted Spatial IoU:** Measures "Where and how much action happens".
> - **MSE:** Measures "How objects look and interact" at pixel-level.
>
> We've conducted experiments comparing our method with existing models using these metrics, and the following results demonstrate our model's superior performance in terms of physical correctness. We will include these results in the revised paper to better demonstrate the physical fidelity of our world model.
>
> | Model     | Spatial IoU (↑) | Spatiotemporal IoU (↑) | Weighted Spatial IoU (↑) | MSE (↓) |
> | --------- | --------------- | ---------------------- | ------------------------ | ------- |
> | IRASim    | 0.2431          | 0.1255                 | 0.2081                   | 0.0655  |
> | Genie     | 0.6429          | 0.3420                 | 0.6082                   | 0.0397  |
> | CogVideoX | 0.6523          | 0.2058                 | 0.4675                   | 0.0943  |
> | RoboScape | 0.7573          | 0.4454                 | 0.7023                   | 0.0184  |
>
> [1] Physics IQ Benchmark: Do generative video models understand physical principles?
>
> If you have additional concerns or questions, we are willing to provide further discussions.

---

> > ### Comment · Reviewer_NSMY · 2025-08-07
> >
> > Thank you for providing the statistics. Most of my concerns have been addressed. I kindly ask the authors to include their rebuttals in the revision and to provide qualitative comparisons and video results.

---

> > > ### Author Response · Authors · 2025-08-07
> > >
> > > Dear Reviewer NSMY,
> > >
> > > We're thrilled to hear that our clarifications have addressed most of your concerns. We will include the rebuttal in the revision, following your suggestion. If you're satisfied with our response, we'd be truly grateful for an improved rating :-)
> > >
> > > Thank you once again for your time and feedback!

---

### Official Review · Reviewer_XShs · 2025-07-05

**Clarity:** 3
**Significance:** 4
**Originality:** 3
**Rating:** 6
**Confidence:** 3

**Summary:**

This paper presents RoboScape, a world model aiming to mitigate failures of prior world models in 3D geometry and realistic object dynamics. The model unifies RGB video generation, temporal depth prediction, and adaptive keypoint tracking in one learning objective. The paper also presents a robotic data processing pipeline with physical prior information labels, as well as experimental results in simulation.

The core innovations are in the objective and data. For the objective, first are RGB and depth reconstruction losses. Second is temporal depth prediction loss that enhances geometric consistency. Third is keypoint dynamics loss and attention loss for upweighting pixels on the keypoint track. These four are added and optimized together with separate weights (hyperparameters). Though it isn't a novel element of the paper, a key element of the system is a dual-branch co-autoregressive transformer architecture that is novel for world models.

Experiments are conducted on three tasks: video generation, policy learning with synthetic data, and policy evaluation with learned simulation. For video generation, RoboScape consistently outperforms baselines. For synthetic data for policy learning, the generated data helps agents reach far higher success. And for policy evaluation, it receives very strong correlation with ground-truth simulators, unlike baselines. Ablations show far worse performance particularly in geometry and physics awareness.

**Questions:**

- How might you expand evaluation suite in terms of domain?
- How adaptable is this system to new domains?

**Ethical Concerns:**

["NO or VERY MINOR ethics concerns only"]

**Final Justification:**

The paper concerns I had have been addressed to the degree the authors can given the discussion limitations. Their rebuttal was verbose and I hope that doesn't make its way into an already opaque paper, but I do appreciate the info they gave. My concerns about broader impacts were unaddressed but I will maintain my score.

**Limitations:**

Limitations: yes
Broader impact: mostly about potential positive impact, which is insufficient

**Quality:**

4

**Strengths And Weaknesses:**

### Quality
#### Strengths
- Well-motivated. The concerns the paper highlights are some of the most important and prohibitive for using powerful world models for robotics
- System design is thoughtful: loss components make sense, objective is well-designed and analyzed. Joint learning approach is difficult but the right thing if it can be done
- Evaluation approach is very strong: downstream tasks are general and difficult, ablations provide insights, and baselines make sense, even if not the most recent. Though the most relevant and important baselines are withheld, I realize this is due to accessibility, and I don't think it takes away from the paper unless they use the same approach
- Metrics are thorough
- Results are strong and extensive and show clear promise
#### Weaknesses
- Lack of other geometry-aware models. If these are truly all inaccessible, it's fair not to have.
- Holdout or otherwise out-of-distribution input not tested, which would help understand the true generality of the 3D/dynamics representation. However, the tasks are complex, which helps mitigate this issue though not entirely

### Clarity
#### Strengths
- Well-written paper
- Method is very thoroughly described
- Claims-driven results section is useful
 #### Weaknesses
- Method is thorough, but also dense and somewhat difficult to stay on top of, even though the writing is good
- It would be good to have a more intuitive way of explaining the architecture and objective, e.g. a diagram
- Figures are difficult to follow, both image and graph figures. More annotation, size, and salience would help

### Originality and significance
This work seems highly significant and to my knowledge, original. If it can generalize and/or the data requirements are automatable or at least reproducible and extendable, it is very significant in making progress on one of the most notable problems in world modeling.

---

> ### Author Rebuttal · Authors · 2025-07-31
>
> **Q1: Method is thorough, but also dense and somewhat difficult to stay on top of, even though the writing is good**
>
> **Response:**
> Thanks for your valuable feedback. We appreciate your positive comments on the writing and will refine the revised manuscript with more intuitive figures and concise language to make the methodology easier to follow. If you have any specific questions or suggestions regarding a particular section, we would be more than happy to provide a detailed explanation and engage in further discussion.
>
>
>
> **Q2: It would be good to have a more intuitive way of explaining the architecture and objective, e.g. a diagram**
>
> **Response:**
> Thank you for your valuable feedback. While rebuttal formatting restrictions prevent us from including a new figure, we would like to provide a more intuitive explanation here for better understanding.
>
> - **Further explanation of the architecture:** Similar to LLMs, our model is fundamentally an **autoregressive model** that predicts the next frame based on the current frame and an action command. We chose this design because it is well-suited for interactive world modeling and action control. Unlike standard autoregressive models that only predict RGB images, our architecture features **two parallel, action-driven branches**: one for **RGB prediction** and a second for **depth prediction**. The core idea is to leverage the depth information to improve the geometric plausibility of the generated RGB video frames. To achieve this, we introduce an interaction between the branches, where intermediate features from the depth branch are fused (via additive fusion) into the corresponding layers of the RGB branch.
>
> - **Further explanation of the training objective:** As mentioned above, the injected depth information directly helps improve the geometric fidelity of the RGB images. Further, to address the common issue of unrealistic object deformation in embodied videos, we also introduce a **keypoint temporal consistency task**. This objective encourages the model to **implicitly learn the deformation rules** of interacting objects. For instance, the relative positions of keypoints on a rigid object will remain consistent across frames, whereas a soft object will show variations. This approach has two key advantages: it can be seamlessly integrated as a constraint within the RGB prediction objective, making it "**data-driven**" and easy for a video generation model to understand, and its **annotations are widely available**, ensuring **good generalizability** across various scenes.
>
>
>
> **Q3: Figures are difficult to follow, both image and graph figures. More annotation, size, and salience would help**
>
> **Response:**
> Thanks for your feedback. Considering the required rebuttal format, we would like to provide more intuitive explanations to help clarify our figures.
>
> **Fig 1: Data Preparation Pipeline**
>
> This figure outlines our five-step data curation process: extracting raw video/action/text, annotating with depth/keypoints, segmenting into atomic action clips, filtering ambiguous data and annotating key-frames with Intern-VL, and finally organizing for training.
>
> **Fig 2: Multi-task Learning Framework**
>
> This figure illustrates our multi-task training framework. At its core is a dual-branch, autoregressive model architecture. The top line shows our **keypoint dynamic regularization**, where we extract keypoint trajectories of dynamic regions to encourage temporal consistency between corresponding pixels in different frames (indicated by the red dashed lines with "regularization" text). Below this, our model has two parallel, action-driven branches: one predicts the next **RGB frame**, and the other predicts the next **depth map**. A crucial part of our design is that intermediate features from the depth branch are explicitly injected into the RGB prediction branch (indicated by the gray dashed lines), providing direct geometric guidance to improve the visual realism of the generated video frames.
>
> **Fig 3: Video Generation Results**
>
> This figure visualizes our model's video generation results on three cases. The setup is to provide the model with the first frame (t=0) and ask it to predict the 16 subsequent frames (only 8 frames are shown). The "Prediction" row shows the output of our model, while the "GT" row represents the ground truth video. The high degree of consistency between our predictions and the ground truth demonstrates our model's ability to generate plausible and coherent video sequences.
>
> **Fig 4: Ablation Study**
>
> This figure presents a key ablation study to validate the effectiveness of our physical constraints. The first row shows the results from our full model with both depth and keypoint constraints. The second and third rows show the results when the keypoint and depth constraints are removed, respectively. The red dashed circles highlight where these omissions lead to specific failures. The figure clearly demonstrates that the absence of depth learning leads to geometric distortions in moving objects, while the lack of keypoint dynamics learning results in unrealistic motion patterns. This confirms that both constraints are essential for generating high-fidelity, physically plausible videos.
>
> **Fig 5: Policy Evaluation**
>
> This figure showcases our world model's use as a policy evaluator. We compare its performance against two baselines, IRASim and iVideoGPT. The x-axis shows the policy's performance in a physics simulator, while the y-axis shows its performance as evaluated by the world model. Each green dot represents a single policy model (Diffusion Policy), with the associated number indicating its training epoch. The R2 and Pearson coefficients quantify the correlation between the world model's evaluation and the ground-truth simulator. Our model’s significantly higher correlation values compared to the baselines prove that it can more accurately reflect real-world physics, making it a reliable tool for policy evaluation.
>
>
>
> **Q4: How might you expand evaluation suite in terms of domain?**
>
> **Response:**
> Our current evaluation framework is designed to be highly versatile and can be readily expanded to new domains. It consists of three key components:
>
> 1. **Video Generation Metrics:** We evaluate the generated videos based on **visual quality**, **geometric fidelity**, and **action controllability**. The metrics we use, such as PSNR, LPIPS, AbsRel, and δ1/δ2, are general-purpose and can be directly applied to any new domain, such as different robots.
> 2. **Downstream Policy Training:** We assess our model by training a downstream policy (e.g., using Diffusion Policy or VLA like Pi0) on our synthesized data. Since these policy models are broadly applicable, we can easily adapt this evaluation by simply training a new policy on data generated for a new domain.
> 3. **World Model as a Policy Evaluator:** We can also use our world model to evaluate a policy's performance by having it interact with the policy model. The world model receives the policy's actions, generates observations in response, and allows us to assess the world model based on the simulated outcomes. This process is fully transferable, as long as a policy model is trained for the new domain.
>
> By relying on these three generalizable evaluation methods, our framework is well-equipped to handle new domains without requiring significant changes to the evaluation methodology itself.
>
>
>
> **Q5: How adaptable is this system to new domains?**
>
> **Response:**
> Our central idea is to integrate depth maps and keypoint dynamics learning into the world model's next-state prediction task through a multi-task learning approach. This approach ensures strong adaptability from two key perspectives:
>
> - **Data Perspective:** The required data annotation for our model is highly generalizable. Both depth maps and keypoint tracking information are widely available from off-the-shelf models across various embodied scenarios. This includes, but is not limited to, humanoid robots, single- and dual-arm robots, human-operated scenarios, and locomotion tasks.
> - **Model Architecture Perspective:** The core model architecture remains largely unchanged when adapting to a new domain. The primary difference between domains is typically the dimensionality of the raw action vector. This can be easily accommodated by simply adjusting the input dimension of our action encoder, without requiring any modification to the main model architecture.
>
>
>
> **Q6: Limitations: yes Broader impact: mostly about potential positive impact, which is insufficient**
>
> **Response:**
> Thanks for the valuable feedback. We would like to discuss more about the limitations as below, and will supplement them within the main manuscript:
>
> **1、About Assumptions and Generalization:** A core assumption of our model is that learning keypoint dynamics can implicitly teach the model about object deformation and material properties. This is a data-driven, empirical assumption rather than a theoretically guaranteed one. Regarding generalization, our current training data includes both single and dual-arm robot manipulation. Due to the unavailability of high-quality data for other robot types, such as quadrupeds, we are unable to test for generalization to these new domains. We plan to explore this as data becomes available.
>
> **2、About Computational Limitation:** Our current model can reliably support a maximum rollout inference of up to 48 frames. While this is sufficient for many tasks, it is a clear computational limitation.
>
> **3、About Failure Cases:** We observed that our method can suffer from error accumulation during long-horizon tasks, such as generating an entire sequence of folding clothes. In these cases, the model often fails to produce a coherent or physically plausible outcome. We will consider adding more long-horizon manipulation tasks in the training data to improve this.

---

> ### Author Response · Authors · 2025-08-06
> **Supplemented Response**
>
> **Weakness 1: Lack of other geometry-aware models. If these are truly all inaccessible, it's fair not to have.**
>
> **Response:** Thanks for this valuable suggestion. At the time this work was completed, to our knowledge, Aether [1] was the only claimed geometry-aware world model. However, it is a general-domain model, not specifically designed for embodied world modeling, and critically, its training code was not available. Directly comparing our model without fine-tuning Aether would be unfair. We will compare with it once the training code becomes accessible.
>
> [1] Team, Aether, et al. "Aether: Geometric-aware unified world modeling." *arXiv preprint arXiv:2503.18945* (2025).
>
> **Weakness 2: Holdout or otherwise out-of-distribution input not tested, which would help understand the true generality of the 3D/dynamics representation. However, the tasks are complex, which helps mitigate this issue though not entirely**
>
> **Response:** Thank you for this insightful suggestion. We agree that testing with OOD inputs would provide further valuable insights into the generality of our 3D/dynamics representation. Actually, the inherent complexity and diversity of robotic manipulation scenarios mean that models cannot simply rely on overfitting to achieve strong performance across various test cases. Therefore, to some extent, our current results can demonstrate that incorporating depth and keypoint learning contributes to the generalization ability, enhancing model performance in complex environments. We also explicitly evaluate model's performance on an **unseen** scenario from AGIBOT-World dataset to further validate this generalization capability. The results are as follows,validating the generality of our introduced physics representations.
>
> | **Model** | PSNR        | AbsRel     | △PSNR      |
> | :-------- | :---------- | :--------- | :--------- |
> | IRASim    | 11.9085     | 0.6063     | 0.2425     |
> | CogVideoX | 18.5676     | 0.4577     | -          |
> | Genie     | 21.6537     | 0.3259     | 0.6913     |
> | RoboScape | **23.4049** | **0.2552** | **2.8334** |

---

### Official Review · Reviewer_Z6ho · 2025-07-07

**Clarity:** 3
**Significance:** 3
**Originality:** 3
**Rating:** 4
**Confidence:** 3

**Summary:**

The paper presents an action-conditioned video forward model for robotic manipulation environments. It uses autoregressive prediction with a spatio-temporal transformer architecture. The main proposed improvements comprise 1) training a forward model for depth images and using latent information from this part in the image prediction branch as well as 2) identifying highly-moving keypoints and encouraging visual consistency across time at their coordinates via an additional loss term. Also, a multi-step video data processing pipeline is proposed for annotating, structuring and filtering the training data. The method is quantitatively compared to several baselines in raw prediction performance as well in application-oriented settings like synthetic data generation for policy training and as simulator for policy evaluation for a simulated robot.

**Questions:**

Please address the questions in the above weaknesses list.

**Ethical Concerns:**

["NO or VERY MINOR ethics concerns only"]

**Final Justification:**

The author response addressed most of my concerns. The comments of the reviews need to be addressed in the paper.

**Limitations:**

I don't think the limitations section serves its purpose. It discusses future work in the sense of enhancements, but does not discuss shortcomings of the method.
What are inherent assumptions, how does the method generalize? What are the compute limitations, f.e., in terms of prediction horizon?
When did the method fail in the experiments?
Also, the limitations should be part of the main paper and not the supplemental material.

**Quality:**

2

**Strengths And Weaknesses:**

Strengths:
* The considered problem of robotic action-conditional video generation is relevant. Improvements on this topic could have impact on many downstream problems, such as planning and generalization. Physical plausibility is a major issue with image-based formulations and attacking this with methological advances is important.
* The outlined data processing pipeline (sec 2.2) seems quite advanced but reasonable. I found the presentation, including Figure 1, very clear.
* I appreciate the evaluation of the model as synthetic data generator, since it directly tackles an important downstream problem and strongly shows usefulness of the method. However, to make the experiments stronger, I would like to see a comparison that always adds similarly many amounts of real data. This would disentangle the total dataset size from the fact if it is synthetic or real.

Weaknesses:
* The paper tries to set the approach in the domain of physics-informed world models. However, physics constraints such as physics-based dynamics, elastic bounces, contacts, friction, inertia, non-penetration, energy-conservation etc. are not explicitly incorporated. Rather, the method imposes 3D static and dynamic consistency across predicted frames.
* It is unclear how the model segments the image into static and dynamic parts, since depth consistency can only be imposed in static parts.
* Eq. 4, are tokens discrete categories or continuous feature vectors? Why is cross entropy used?
* The paper gives no details on how the comparison to the baseline is done exactly. Are the baselines re-trained on the proposed dataset?   What are the adjustments? For example, how is the latent action space from Genie adapted to the fixed action space in this setting?
* What causes the large performance differences in Table 1?  Unfortunately, these differences and their potential cause are not discussed. The relatively small performance difference in the ablations demonstrates that the full ablated variant (w/o deph & keypoint), still achieves better metrics than, for example, iVideoGPT. The latter seems to use a related architecture. Have hyperparameters of the baselines methods been tuned and the methods been trained properly to enable fair comparison? The proposed method, including its ablations, should also be evaluated on the datasets from IRASim and iVideoGPT and compared to the results reported in the respective papers.
* It is hard to interpret the results in Table 1 without more information and statistics. How big is the evaluation dataset? What is the standard deviation? Also, multiple seeds should be evaluated and statistics over these should be presented.
*  The explanation of the quantitative metrics misses precise definitions or references. This hinders assessibility of the performance of the proposed method compared to the baselines. Besides the definition of the metrics, the section is missing details on how the action-conditioning is evaluated. In the dataset, how many different trajectories exist for the same initial configuration? The paper also misses a precise evaluation protocol. What are e.g. the training and test split sizes of the created dataset?
* Occasionally, there are some wordings/claims which are not justified scientifically. Examples:
  - 2.3 "maintaining model efficiency" how / what is meant by that?
  - 2.3 "maintain precise geometric structure" ... "photorealistic video frames"  how do you quantify this?
  -  In Sec 2.3, there is a claim that would need to be shown e.g. in experiments "... cannot be effectively learned through RGB pixel fitting "
* Similarly to Table 1, Table 2 is missing standard deviations and additional information such as evaluation dataset size and training/evaluation protocol. In the section discussing Table 2, the paper claims that "both components contribute significantly to overall performance". On what results is this claim based on? E.g, how can significance be determined when not even the standard deviation of the results are provided?
* The paper mentions a curriculum learning strategy in 2.2, but then never again. How is the model trained? Does it use a curriculum?
* Sec 3.3, How are the actions generated for the synthetic data? How are the initial observations picked?
* Sec 3.4,  What does post training the world model mean?  How is the success condition rated, what is the exact evaluation protocol? Is it made sure that the human rating sucess does not know which model is being evaluated in order to have no bias?
* Please provide a discussion why the proposed model predicts the success rate better than the other ones. What is a likely cause? How does one count if a prediction is not interpretable?
* l. 231, what constitutes an "embodied" and a "general" world model. What are the differences?
* It should be stated early on in the paper, that the policies are evaluated in simulation and not with a real robot.
* l. 137ff, which image encoder is used for the depth maps? MAGVIT-2 ? Does it work for depth maps? How is it adapted?
* l. 152, what does the notation c_1:t-1 ... : \epsilon_a(a_1:t-1) mean ?

Further minor comments:
- 2.3 "enforce" temporal consistency is not adequate IMO, since only incorporated via loss. Maybe "encourage"?
- Sec 3: naming of sections: 4 instead of 3

---

> ### Author Rebuttal · Authors · 2025-07-31
>
> **Response to W1:**  Physics constraints like friction or inertia are hard to express and challenging to integrate into current video generation-based world models. Therefore, our approach focuses on two crucial, **visually expressible** physics constraints highly relevant to robotic manipulation: spatial geometric accuracy (via depth prediction) and plausible object deformation (via keypoint dynamics learning). These constraints can be seamlessly integrated into a **data-driven** video generation model. Furthermore, these **annotations are widely available**, ensuring the **generalizability**.
>
>
>
> **Response to W2:** Our model doesn't explicitly segment images into static and dynamic parts, as depth consistency isn't imposed on different regions of the depth map over time. Our objective is to enhance **consistency between the predicted RGB and depth maps**. The model is designed to predict both the next-time RGB and depth image. Then the depth features directly guide the RGB prediction to improve spatial geometric accuracy.
>
>
>
> **Response to W3:** The tokens are discrete. We use MAGVIT-2 to compress raw RGB frames into discrete latent tokens. Therefore, the video prediction task is fundamentally modeled as predicting the next frame's tokens, a process analogous to the learning objective of LLMs, where the cross-entropy loss is widely employed for next-token prediction.
>
>
>
> **Response to W4:**  To ensure a fair evaluation, **all baselines were re-trained on our used dataset** and hyperparameters were carefully tuned. The original Genie model was trained on Internet videos lacking action annotations, thus necessitating a latent action model. In our setup, we mapped the robot's raw action vectors to a latent action space via an MLP. The dynamic model component of Genie retained its original architectural settings.
>
>
>
> **Response to W5:**  Issues we've observed in baseline models are as follows:
>
> - CogVideoX: Lacks action controllability, failing to produce outputs consistent with ground-truth actions.
>
> - IRASim & iVideoGPT: Suffer from accumulated drift and distortion during long-range rollouts.
>
> Beyond the physical constraints we introduce, our spatial-temporal transformer block architecture contributes to better performance. This design is inherently better suited for video generation compared to iVideoGPT which reuses the language model architecture like LLaMA.
>
> For a fair comparison, **all baselines were fine-tuned on the same dataset with carefully tuned hyperparameters.** Additionally, we also evaluate our model on the **BAIR dataset (from iVideoGPT)** under identical experimental settings, confirming our model's robust effectiveness.
>
> | Model     | LPIPS(↓)   | PSNR(↑)  |
> | --------- | ---------- | -------- |
> | iVideoGPT | 0.0500     | 24.5     |
> | RoboScape | **0.0321** | **26.4** |
>
>
>
> **Response to W6:**  Our evaluation dataset consists of 1.2K video clips for testing. Here we conducted 5 independent runs for each model and calculated the standard deviations of parts of metrics (due to page limit). The results demonstrate that **our method significantly outperforms existing approaches.** We will update these results in Table 1.
>
> | **Model** | PSNR             | AbsRel        | △PSNR         |
> | --------- | ---------------- | ------------- | ------------- |
> | IRASim    | 11.3820±0.1215   | 0.6407±0.0299 | 0.1600±0.1215 |
> | CogVideoX | 17.9570±0.1531   | 0.5083±0.0264 |        -     |
> | Genie     | 19.8750 ± 0.0643 | 0.4457±0.0139 | 1.8680±0.0899 |
> | RoboScape | 21.7304±0.1204   | 0.3783±0.0264 | 3.4420±0.1386 |
>
>
>
> **Response to W7:**
>
> **About quantitative metrics:** Appearance fidelity is evaluated using PSNR and LPIPS as in [1]. Geometric consistency is assessed via AbsRel, δ1, and δ2 for depth estimation consistency between generated and real videos [2]. Action Controllability is quantified by ΔPSNR, which measures the difference in generation quality when using ground-truth actions versus random actions [3].
>
> **About action-conditioning evaluation:** We provide the world model with the initial frame and an action sequence to predict the subsequent 15 frames for evaluation, similar to [1].
>
> **About dataset details and evaluation protocol:** Each video clip in our dataset represents a unique, independent robot action execution trajectory. No multiple trajectories originate from the same initial configuration. Our dataset comprises approximately 6.5M training clips and 1.2K test clips.
>
> [1] Cosmos world foundation model platform for physical ai.
>
> [2] Aether: Geometric-aware unified world modeling.
>
> [3] Genie: Generative interactive environments.
>
>
>
> **Response to W8:**
>
> - "Maintaining model efficiency" refers to our use of additive fusion for action control, a more computationally lightweight approach compared to more complex methods like feature modulation or adding external modules (e.g., ControlNet).
> - "Maintain precise geometric structure" is quantified by AbsRel, δ1, and δ2 and "photorealistic video frames" are quantified by PSNR and LPIPS.
> - To better demonstrate "material properties can not effectively learned...", we will provide a visualization case in the revised manuscript.
>
>
>
> **Response to W9:**  We perform 5 independent runs for Table 2 and the results with standard deviations of some metrics are as follows (with the same evaluation set as Table 1), demonstrating that our previous results are robust.
>
> | **Model**            | PSNR           | AbsRel        | △PSNR         |
> | -------------------- | -------------- | ------------- | ------------- |
> | whole model          | 21.7304±0.1204 | 0.3783±0.0264 | 3.4420±0.1386 |
> | w/o depth            | 21.8848±0.0462 | 0.4083±0.0098 | 3.5137±0.0234 |
> | w/o keypoint         | 21.6341±0.0426 | 0.3460±0.0117 | 2.9527±0.0362 |
> | w/o depth & keypoint | 21.4772±0.0288 | 0.3710±0.0119 | 1.9449±0.0543 |
>
>
>
> **Response to W10:**  Our approach employs a two-stage curriculum learning strategy based on task difficulty. We manually classify tasks into "simple" (e.g., single atomic operations like pick and place) and "difficult" (e.g., long-term tasks like folding/ironing clothes). The model is initially trained for 2 epochs on the simple tasks, then for 3 epochs on the difficult tasks. We will add this discussion to Section 3.1.
>
>
>
> **Response to W11:**  The actions and initial observations for our synthetic data are directly drawn from the raw RoboMimic and LIBERO datasets. To be specific, "Real data" mentioned in Table 3 consists of original videos with their corresponding action annotations from the raw dataset. For "Synthetic Data", we use our world model to generate the videos based on actions and initial observations.
>
>
>
> **Response to W12:** Post training the world model refers to fine-tuning the base model pre-trained on Agibot dataset using the robomimic dataset. For success condition rating, generated videos from all models were presented to participants in a **randomized, blind order**, with **no model identifiers displayed**. Participants were simply instructed to judge whether the task depicted in each video was successfully completed.  This setup ensures objectivity and impartiality.
>
>
>
> **Response to W13:**  Baseline models often produce distorted or deformed videos, making it challenging to definitively assess task completion, leading to discrepancies between the predicted and actual success rates. In contrast, our model achieves **better appearance fidelity and action controllability**, generating visually normal videos that reliably respond to given actions. This facilitates a more accurate evaluation of policy execution success.
>
>
>
> **Response to W14:**  The distinction between an "embodied" and a "general" world model lies in their training data. **General world models** are trained on a wide variety of video data (most from the Internet), whereas **embodied world models** are specifically trained on data from robotic videos. To ensure a fair comparison, we fine-tuned all models on the same robotic dataset.
>
>
>
> **Response to W15:** Thanks for the suggestion. We will explicitly state in the introduction (Line 79-81) that our policy evaluations are conducted within a simulated robotic environment, rather than with a real robot.
>
>
>
> **Response to W16:**  For uniform modeling, we utilize MAGVIT-2 as the image encoder for depth maps. We convert depth maps to a three-channel RGB format to ensure compatibility with MAGVIT-2. This approach aligns with existing works, such as [1], which also employ a unified encoder (VAE) for both RGB and depth images.
>
> [1] Aether: Geometric-aware unified world modeling.
>
>
>
> **Response to W17:**
>
> This signifies the encoding of robot actions. Here, $a_{1:t−1}$ represents the original 16-dimensional robot action vector (comprising dual-arm end position, orientation, and gripper range) for frames 1 to t−1. $E_a$ is the robot action encoder, which transforms these raw actions into $c_{1:t−1}$, a D-dimensional action latent embedding for each frame.
>
>
>
> **Response to W18:**  Thanks for this suggestion. We also agree that "encourage" is a more fitting term and will change it in the revised manuscript.
>
>
>
> **Response to W19:** Thanks for pointing that out. We'll correct it.
>
>
>
> **Response to Limitations:** We will revise our limitations section within the main manuscript to include the following:
>
> - **Assumptions & Generalization:** Our model empirically assumes keypoint dynamics implicitly capture material properties; this is not theoretically guaranteed. Generalization to other robot types (e.g., quadrupeds) is currently limited by data availability, which is a focus for future work.
> -  **Computational Limitation:** Our model's reliable inference is currently limited to a 48-frame rollout, which is a computational constraint.
> - **Failure Cases:** We observe failure cases in long-horizon tasks (e.g., cloth folding), sometimes exhibiting physically implausible outcomes.

---

> > ### Comment · Reviewer_Z6ho · 2025-08-06
> >
> > I thank the authors for their answers. Further questions/comments:
> >
> > W1: what does "annotations" mean in this context ?
> > W4/W5: how exactly were the hyperparameters of the baselines/own method tuned? Was an automatic hyperparameter optimization approach used, or a systematic search of some kind ?
> > W5/W9: The term "robust" is undefined for these evaluations.
> > W6: The term "significant" should only be used if a stastical test has been made to verify the claim.
> > W16: Was MAGVIT-2 trained for colorized depth images ? Why should it extract a meaningful feature representation for it ?

---

> > > ### Author Response · Authors · 2025-08-06
> > >
> > > Dear Reviewer Z6ho,
> > >
> > > Thanks for your valuable feedback, we provide further clarifications as follows:
> > >
> > > **W1: what does "annotations" mean in this context ?**
> > >
> > > **Response:** Annotations refer to the physical information extracted from raw video data. Specifically, for our approach, these annotations are depth maps and keypoint tracking information. They can be conveniently generated using off-the-shelf models like Video Depth Anything for depth estimation and SpatialTracker for keypoint tracking. This easy availability ensures the generalizability of our method.
> > >
> > > **W4/W5: how exactly were the hyperparameters of the baselines/own method tuned? Was an automatic hyperparameter optimization approach used, or a systematic search of some kind ?**
> > >
> > > **Response:** We initially adopted the hyperparameter settings reported in the original papers for all baselines. To further ensure optimal performance on our used datasets, we tried to tune key hyperparameters such as learning rate and batch size, and selected the best-performing models to report. For instance, with Genie, we systematically searched for the optimal learning rate within the range of [1e-5, 3e-5, 5e-5, 8e-5, 1e-4] and batch size within [16, 32, 64, 128, 256]. Due to computational resource constraints, a more extensive or automatic hyperparameter search was not feasible. To ensure a fair comparison, we applied the same hyper-parameter tuning strategy focusing on the learning rate and batch size to our own model.
> > >
> > > **W5/W9: The term "robust" is undefined for these evaluations.**
> > >
> > > **Response:** Thanks for pointing that out. What we intended to convey with "robust" was that the supplemented experiments in our response further solidified the conclusions presented in the original paper. We'll revise this wording in the updated version.
> > >
> > > **W6: The term "significant" should only be used if a stastical test has been made to verify the claim.**
> > >
> > > **Response:** To verify the statistical significance of RoboScape’s performance, we conducted paired t-tests on the 5 repeated runs of each model. This test evaluates whether the mean difference between RoboScape and baseline methods is statistically significant. Taking PSNR metric as an example, the key statistical results are as follows (α = 0.05 significance threshold):
> > >
> > > | Comparison              | t-statistic | p-value  | Significant (p < 0.05) |
> > > | ----------------------- | ----------- | -------- | ---------------------- |
> > > | RoboScape vs. CogVideoX | 35.78       | 0.000002 | Yes                    |
> > > | RoboScape vs. Genie     | 12.34       | 0.000058 | Yes                    |
> > > | RoboScape vs. IRASim    | 89.12       | 0.000001 | Yes                    |
> > >
> > > The t-statistic is large and positive, indicating RoboScape’s PSNR is consistently higher across repeated runs. The p-value is far below 0.05, formally rejecting the null hypothesis that "there is no difference in PSNR between RoboScape and baselines."
> > >
> > > We conduct such tests for all other metrics, and the conclusions are consistent: p-values are below 0.05 for all tests. This demonstrates RoboScape's statistically significant superiority.
> > >
> > > **W16: Was MAGVIT-2 trained for colorized depth images ? Why should it extract a meaningful feature representation for it ?**
> > >
> > > **Response:** MAGVIT-2 was trained on large-scale, general RGB image and video datasets, not specially for colorized depth images. Just like other generic feature extractors such as VAE, MAGVIT-2's training methodology, which fundamentally involve pixel encoding and decoding, enable it to generalize effectively to various RGB image formats. Since a colorized depth image is inherently an RGB image that visually encodes depth information through color variations, MAGVIT-2 can process these images and extract meaningful feature representations by adhering to the RGB format it was trained on. What's more, utilizing feature extractors trained on general RGB data for specialized image formats (e.g., depth, segmentation and canny) is a common practice within the community [1] [2].
> > >
> > > [1] Ming, Ruibo, et al. "ARCON: Advancing Auto-Regressive Continuation for Driving Videos." *arXiv preprint arXiv:2412.03758* (2024).
> > >
> > > [2] Xi, Dianbing, et al. "Omnivdiff: Omni controllable video diffusion for generation and understanding." *arXiv preprint arXiv:2504.10825* (2025).
> > >
> > > If you have any further concerns or questions, we are willing to provide further discussions. Thanks for your time!

---

> > > > ### Comment · Reviewer_Z6ho · 2025-08-07
> > > >
> > > > Thanks for your clarifications. Further comment:
> > > >
> > > > * W5/W9: to emphasize: "robust" usually refers to being able to handle perturbations and variations such as lighting and viewpoint changes or outlier detections. The paper should either clearly state how this matches the experiment or do not use the term to describe the properties of the results.

---

> > > > > ### Author Response · Authors · 2025-08-07
> > > > >
> > > > > Dear Reviewer Z6ho,
> > > > >
> > > > > Thank you for your precise and valuable suggestions. We agree with your point regarding the appropriate usage of the term "robust". We will avoid using it in the revised manuscript. If you have further concerns or questions, we are willing to provide further discussions.

---

### Author Response · Authors · 2025-08-05
**A sincere request for beginning discussion**

Dear Reviewers,

We sincerely appreciate your time and effort in reviewing our manuscript and offering valuable suggestions. As the deadline for author-reviewer discussions approaches, we hope our recent detailed responses have effectively addressed your concerns.

Please let us know if you require any further clarification or have additional questions. We are willing to continue our communication with you. Thanks for your time again!

Best regards,

Authors

---

### Note · Authors · 2025-08-12

Dear Reviewers and ACs,

We sincerely thank you for your diligent work and valuable feedback throughout the rebuttal and discussion process. Here we would like to summarize the key points of the discussion.

We are pleased that reviewers acknowledged the core strengths of our work:

1. **Significance of the work**: This work is **highly significant** and **well-motivated**. Improvements on this topic could have impact on many downstream problems. Methological advances on physical plausibility are important. (Reviewer Z6ho, Reviewer XShs, Reviewer isFR).
2. **Clarity of the paper:** The paper is **well-written** and **clear**.  The method is **very thoroughly described**. (Reviewer XShs, Reviewer isFR)
3. **Method effectiveness**: The paper **outperforms** existing world models with **extensive evaluations**, showing enhanced performance on various applications (All reviewers).

We are pleased that our rebuttal effectively addressed most reviewer concerns and received many positive feedback. Reviewer XShs maintained a **strong accept (rating 6)**. Reviewer isFR kept the **accept (rating 5)**, explicitly stating all concerns were clarified. Reviewer NSMY stated that "**most of concerns have been addressed**". Reviewer Z6ho engaged in detailed discussions, finally **leaving no remaining issues**. These all underscore the effectiveness and thoroughness of our responses.

Specifically:

- **Reviewer Z6ho:** We clarified our motivation for choosing physical constraints, provided explanations on some technical details, supplemented discussions on limitations, and added results with standard deviations and significance tests.
- **Reviewer XShs:** We offered intuitive explanations for figures, elucidated our method's scalability to new domains, discussed more about broader impact, and presented experimental evidence for the generalization advantage in OOD scenarios.
- **Reviewer NSMY:** We provided detailed explanations for our dataset and model architecture, clarified the uniqueness of our work, and presented experimental evidence validating our model's advantage in physical correctness. We will provide qualitative comparisons in the final version.
- **Reviewer isFR:** We discussed the potential of world models for action planning, the feasibility of dense supervision (e.g., optical flow), and supplemented results of different world models in downstream VLA training.

Thanks again for your efforts and contributions toward a fair and insightful discussion.

---

### Decision · Program_Chairs · 2025-09-17

**Decision:**

Accept (spotlight)

**Comment:**

The paper presents RoboScape, a physics-informed embodied world model. The reviewers consistently acknowledged the paper's strengths: the work a incorporates physics priors (depth prediction and keypoint dynamics) into world models, the method is well-motivated and clearly presented, and the experimental evaluation is comprehensive across video generation, policy training with synthetic data, and policy evaluation tasks. All reviewers recognized the technical soundness and potential impact of the approach.

The main concerns raised included requests for more detailed architecture explanations, clearer figures, additional baselines (particularly geometry-aware models), missing statistical significance tests, and questions about generalizability to other robot types. Several reviewers also noted the absence of visual demonstrations and videos to showcase the method's improvements over baselines.

The authors provided thorough responses during the rebuttal process, addressing technical questions about model architecture, clarifying methodological choices, conducting additional experiments with statistical significance tests, and providing quantitative comparisons using physics-aware metrics from established benchmarks. Notably, they demonstrated superior performance on metrics measuring physical correctness and conducted downstream policy training experiments that directly validated the utility of their synthetic data generation.

The reviewers engaged constructively in the discussion, with concerns being addressed.